# Towards fair decentralized benchmarking of healthcare AI algorithms with the Federated Tumor Segmentation (FeTS) challenge

Computational competitions are the standard for benchmarking medical image analysis algorithms, but they typically use small curated test datasets acquired at a few centers, leaving a gap to the reality of diverse multicentric patient data. To this end, the Federated Tumor Segmentation (FeTS) Challenge represents the paradigm for real-world algorithmic performance evaluation. The FeTS challenge is a competition to benchmark (i) federated learning aggregation algorithms and (ii) state-of-the-art segmentation algorithms, across multiple international sites. Weight aggregation and client selection techniques were compared using a multicentric brain tumor dataset in realistic federated learning simulations, yielding benefits for adaptive weight aggregation, and efficiency gains through client sampling. Quantitative performance evaluation of state-of-the-art segmentation algorithms on data distributed internationally across 32 institutions yielded good generalization on average, albeit the worst-case performance revealed data-specific modes of failure. Similar multi-site setups can help validate the real-world utility of healthcare AI algorithms in the future.

Glioblastomas are arguably the most common, aggressive, and heterogeneous adult brain tumors. Despite the proliferation of multimodal treatment composed of maximal safe surgical resection, radiation, and chemotherapy, the median survival is approximately 8 months, with less than 7% of patients surviving for over 5 years[1]. This poor prognosis is largely on account of the pathological heterogeneity inherently present in glioblastomas, leading to treatment resistance, and thus grim patient outcomes. Radiologic imaging (i.e., magnetic resonance imaging (MRI)) is the modality of choice for routine clinical diagnosis and response assessment in glioblastoma patients, and delineation of the tumor sub-regions is the first step towards any computational analysis that can enable personalized diagnostics[2].

While manual annotation is arduous because of the tumor heterogeneity, significant progress has been made in the field of automatic segmentation of brain tumors[3–5]. Translating these research results to real-life applications, however, remains an open challenge, as deep learning models struggle to maintain robust performance in unseen hospitals, if their data was acquired from different imaging devices and populations than the data for model development[6–10]. This can be partially addressed by collecting diverse data centrally to train a robust model that will generate acceptable results on unseen data. However, this centralized data collection is hampered by various cultural, ownership, and regulatory concerns like the Health Insurance Portability and Accountability Act (HIPAA) of the United States and the General Data Protection Regulation (GDPR) of the European Union that restrict data sharing among institutions.

Federated learning (FL)[11] is a promising approach to train robust and generalizable models by leveraging the collective knowledge from multiple institutions, while sharing only model updates with a central server after local training to preserve privacy[12]. In the typical FL workflow, local training at federated collaborators is performed repeatedly in multiple federated rounds, and at the end of each round, the central server aggregates all received model updates into a global model, which is used as the initialization for the next round of federated training. Hence, aggregation methods are a crucial technical aspect of FL and an active field of research[13,14]. The pioneering FedAvg

✉ e-mail: spbakas@iu.edu

aggregation method[11] uses weighted averaging of the updated model parameters from each institution, where the weights are proportional to the dataset size of each site. Building on top of this method, Briggs et al.[15] formulated a strategy of hierarchical clustering that groups sites based on the similarity of local updates and then builds specialized models to better handle data heterogeneity. Their results showcased faster convergence, with substantial differences in the most heterogeneous settings compared to FedAvg. Another study showed how data heterogeneity negatively affects convergence by introducing a drift in local updates[16]. Their approach corrects the introduced drift through variance reduction, resulting in fewer communication rounds and more stable convergence. Although benchmarks for FL methods exist, both for natural images[17] and medical datasets[18,19], only a single, concurrent work[19] follows the design principles of international competitions, also known as challenges[20,21]. These principles require private test datasets for a fair comparison of methods in a continuous evaluation, and equal conditions for all challenge participants. To guarantee equal conditions in the context of FL, it has to be ensured that all algorithms implement FL correctly, in particular avoiding (accidental) data leakage, and that constraints for communication or computation resources are simulated reproducibly.

The central idea of FL−keeping the data distributed and sending around algorithms−is not only a promising avenue for model development, but can also be transferred to a model validation setting. In such a collaborative, multi-site evaluation setting, existing models are shared with clinical data owners for evaluation and the results, including performance metrics and (anonymized) meta-information about the local data, collected for subsequent analysis. This allows validation on datasets that substantially exceed typical test datasets in size and diversity, as clinicians may contribute data without having to publicly release them. Thus, a multi-site evaluation can help to test model robustness and generalizability in the wild, meaning real-world data covering diverse patient population demographics and varying acquisition protocols and equipment. Generalizing to distribution shifts at test time is sometimes referred to as domain generalization, and numerous approaches to this problem have been studied[22]. To measure methodological progress in model robustness, several benchmarks were proposed recently, which evaluate algorithms on test datasets with shifts induced by synthetic image transformations[23], various real-world applications[24], and multi-centric medical datasets[18]. Competitions with realistic shifts between training and test distribution have so far been restricted to small-scale evaluations on a few unseen domains[25,26]. Although multi-site evaluation has been used before in FL studies[27–30], its usefulness to benchmarking independently of FL has only recently been explored[31], and no large-scale multicentric results have been reported for challenges so far.

The rising interest of numerous studies on FL in healthcare[27–30,32–34] highlighted the need for a common dataset and a fair benchmarking environment to evaluate both aggregation approaches and model generalizability. To this end, we introduced the Federated Tumor Segmentation (FeTS) challenge. The primary technical objectives of the FeTS challenge were:

1. Fair comparison of federated aggregation methods: Provide a common benchmarking environment for standardized quantitative performance evaluation of FL algorithms, using multicentric data and realistic FL conditions.
2. Algorithmic generalizability assessment at scale: Evaluating the robustness and generalizability of state-of-the-art algorithms requires large-scale real-world imaging data, acquired at clinical environments from diverse sites. A collaborative, multi-site evaluation approach can assess practical applicability in real-world scenarios.

These goals were reflected in two independent challenge tasks: Task 1 focused on the methodological challenge of model aggregation for FL in the context of tumor segmentation. The primary research goal here was to push the limits of FL performance by innovating on the aggregation algorithm. Additionally, we evaluated whether tumor segmentation performance can be improved while also reducing the federated training time by selecting a subset of collaborators for local training. In Task 2, the objective was to develop methods that enhance the robustness of segmentation algorithms when faced with realistic dataset shifts. We investigated whether brain tumor segmentation can be considered solved in real-world scenarios, and studied the pitfalls associated with collaborative, multi-site evaluation for biomedical challenges, along with potential strategies to address them. To benchmark the best possible algorithms, FL was not a requirement in training models for Task 2. An overview of the challenge concept is given in Fig. 1.

In this work, we present the analysis of the FeTS Challenge results and insights gained during the challenge organization. The contributions of our work are threefold: (1) We introduce a fair and common benchmarking environment for evaluating technical solutions in the context of FL: The FeTS Challenge Task 1 establishes a standardized evaluation framework for comparing federated aggregation methods, assessing their impact on tumor segmentation performance in FL simulations with data from 23 medical sites. This contribution sets the stage for a more accurate and reliable evaluation of FL models in the field. (2) We demonstrate how the biomedical competition format can close the gap between research and clinical application: Unlike previous benchmarks or challenges that relied on small test sets or simulated real-world conditions, the FeTS challenge Task 2 presents an in-the-wild benchmarking approach that evaluates the accuracy and investigates failure cases of segmentation algorithms on a large-scale. We circulate the solutions provided by challenge participants across multiple collaborating healthcare sites of the largest to-date real-world federation[28], replicating real-world conditions during evaluation. (3) We find in Task 1 that adaptive aggregation algorithms and selective client sampling improve the performance of tumor segmentation models. The collaborative, multi-site validation study in Task 2 reveals that these models generalize well on many testing institutions, but their performance drops on others. This suggests that current algorithms may not be robust enough for widespread deployment without institution-specific adaptation.

## Results

The FeTS challenge is a demonstration of international collaboration for algorithmic benchmarking towards highlighting the impact and relevance of methodological innovation: Our dataset comprised contributions from data providers in 17 countries around the globe (Fig. 1), enabling a diverse and comprehensive collection of samples. After its first instantiation in 2021, the FeTS challenge was repeated in 2022 with a more consolidated setup and extended data. We focus on the year 2022 in the main part as the testing data size was much larger than in 2021, but the findings are overall in line (results from 2021 are in the Supplementary Note 5).

While the Brain tumor segmentation (BraTS) 2021 challenge[3] already accumulated a large, multicentric dataset, the collaborative multi-site evaluation (Task 2) in the FeTS challenge further increases the size and diversity of the test set. Data from 24 de-centralized institutions unseen during training were added to 8 institutions from the BraTS challenge test set, which resulted in the inclusion of three additional continents and scaled up the total number of test cases by more than a factor of four. The challenge garnered participation from teams across the globe, attesting to the worldwide interest and engagement in the field of FL in healthcare. Our organizing team was geographically dispersed across three continents, too, embodying the collaborative spirit of this international effort. Specifically, in 2022, the challenge attracted 35 registered teams in total, among whom 7 teams successfully submitted valid entries for Task 1, while 5 teams made

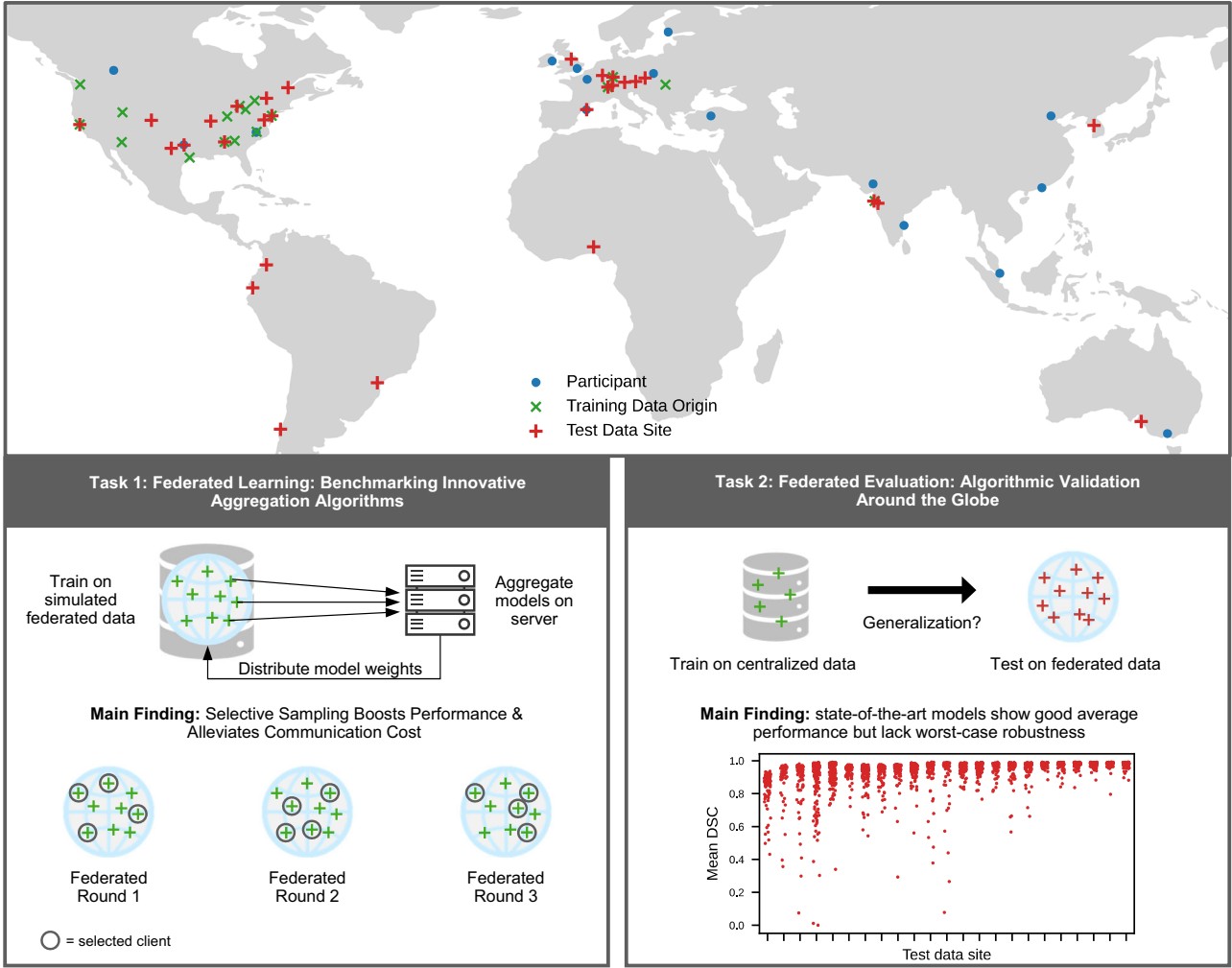

**Fig. 1 | Concept and main findings of the Federated Tumor Segmentation (FeTS) Challenge.** The FeTS challenge is an international competition to benchmark brain tumor segmentation algorithms, involving data contributors, participants, and organizers across the globe. Test data hubs are geographically distributed while training data is centralized. Participants include those from the 2021 and 2022 challenges. Task 1 focused on simulated federated learning and we consistently saw an increase in performance by teams utilizing variants of selective sampling in their federated aggregation. In Task 2, submissions are distributed among the test data hubs for evaluation. As a representative example, the top-ranked model shows good average segmentation performance (measured by the Dice Similarity coefficient, DSC) but also failures for individual cases. Cases with empty tumor regions and data sites with less than 40 cases are not shown in the strip plot. Source data are provided as a Source Data file.

contributions for Task 2. For Task 2, we additionally evaluated 36 more models that had been submitted originally to the BraTS 2021 challenge, as this challenge used the same training images, albeit without the information about institution partitioning (described in the methods section).

## Selective collaborator sampling improves efficiency and performance

The combined results from the simulated FL experiments performed by all participants for Task 1 provided valuable insights into FL methods that improve the efficiency of the federated algorithm while also enhancing the overall segmentation performance, disproving the initial assumption that these two objectives might negatively impact each other. In particular, the natural limitation of the simulated FL experiment time for Task 1 led the participants to explore ideas on how to select collaborators for which to perform local training in each federated round. Training is in general as fast as the slowest collaborator, so the ideas here were based on the question: How do we handle clients with long FL round times? In this challenge, simulated time was the largest for clients with many samples, as their total time is dominated by local training duration, making the time required for

transmitting model parameters negligible in comparison. Large clients hence, play a double role in the Task 1 experiments, as they take the most time but may also aid convergence through many local optimization steps on their rich data.

This dichotomy is reflected in the independent analyses manuscripts of the challenge participants[35–49]. While some teams experimented with dropping slow collaborators[35,38,43,45], they also found that alternating between full participation and dropping slow clients can be a beneficial compromise, which guarantees that all available data are seen. Other teams focused on training on the largest clients[36], arguing that overfitting is less likely on those. Independent of the exact strategy, all teams using selective client sampling consistently reported that it benefits convergence speed without damaging performance and in some cases even improving it. A possible explanation is that in probabilistic sampling methods, the contribution of sites with small datasets is uplifted while they are overwhelmed by big sites in the baseline algorithm that always selects all sites for training. Submissions that used selective collaborator sampling[36,38,43,45] also landed among the top positions in the Task 1 leaderboard (Table 1). Although other algorithm components like the aggregation method also influence the ranking, this trend

**Table 1 | Algorithm characteristics and mean ranking scores of Task 1 submissions**

| Team | Aggregation method | | | | | lr schedule | Client selection | Score |
|---|---|---|---|---|---|---|---|---|
| | DS | PD | LO | LI | Combination | | | |
| FLSTAR | ✓ | | ✓ | | ⊙ | Constant | 6 largest | 2.75 |
| Sanctuary | ✓ | ✓ | ✓ | | ⊙ | Polynomial | Alternating: all; drop slow clients | 3.05 |
| RoFL | ✓ | ✓ | | | ⊙ + server optimizer | Step | All | 3.35 |
| gauravsingh | ✓ | | | ✓ | ⊕ | Constant | 6 random | 3.67 |
| rigg | ✓ | | ✓ | ✓ | ⊕ (weighted) | Constant | Randomly drop large clients | 4.65 |
| HT-TUAS | ✓ | ✓ | | | ⊕ | Constant | 4 random | 4.69 |
| Flair | ✓ | | | | Multiple gradient descent with contraint | Constant | All | 5.85 |

Algorithm characteristics include the aggregation method, learning rate (lr) schedule, and client selection. Algorithms are listed in the order of ranking score contained in the Score column, with the best on top. See the methods section for how the ranking score is calculated. A common pattern for aggregation methods is to compute multiple normalized weight terms (*DS* Dataset size, *PD* (inverse) Parameter distance, *LO* Potential for local optimization, *LI* Local improvement) and combine them either through arithmetic mean (⊕) or multiplicative averaging (⊙). The weight term abbreviations were introduced here as categories summarizing the main idea behind the weight terms, but the implementation details in the teams' algorithms differed slightly, as described in the methods section. Only one team chose a completely different aggregation approach (Flair). Selectively sampling clients was used by five teams to improve the convergence speed.

showcases that these methods hold promise for simultaneously improving convergence speed and performance.

## Adaptive aggregation methods boost performance

In the context of Task 1, aggregation methods take the local model updates from all clients that participated in the last federated training round as input and compute a set of global model parameters from them. Among the algorithms developed by participants in 2022, six of seven were diverse variants of the following high-level approach: (1) compute multiple normalized weighting terms for each collaborator; (2) combine these terms using either additive or multiplicative averaging; (3) output the average of all local models weighted by the combined term of step 2.

Most efforts from the challenge participants concentrated on steps 1 and 2, and only two teams[37,38] also experimented with step 3, by introducing adaptive optimization at the central server[50]. The most popular weighting term (step 1) was proportional to the local dataset size, as proposed in the FedAvg algorithm[11]. Beyond this simple baseline, approaches that adapted the weighting based on the training history (e.g., validation loss of the last round) or based on the inverse parameter-space distance to the average model were explored. Experiments in the independent analyses of the participants showed that some of these adaptive aggregation terms could outperform FedAvg[36–38,42,49], but due to the heterogeneity of experimental setups, there is not a single method that stood out. Combining multiple weighting terms (step 2) proved beneficial for most teams, especially combining the FedAvg term with adaptive terms. In the official challenge results (Table 1, with details for individual evaluation metrics in Supplementary Note 2), methods that combine weighting terms through multiplication (with subsequent normalization) obtained better ranking scores, which is a trend that was also found by one team in experiments for the FeTS challenge[36]. In conclusion, the combined results of experiments performed by challenge participants and the official challenge results produced a variety of methods that adapt the influence of individual collaborators during training to aggregate locally trained models more effectively.

## Multi-site validation reveals mixed generalization

To investigate the influence of data characteristics and algorithmic choices on segmentation performance in the wild, we conducted a collaborative, multi-site evaluation (challenge Task 2). This evaluation encompassed 41 models, which were trained in a centralized fashion and deployed on cases from 32 institutions (also referred to as sites) spanning six continents. Technical issues during the multi-site evaluation caused 5 institutions to run only a subset of models; details on

this are described in the next section. Our analysis revealed substantial performance variations among different sites, with certain institutions also exhibiting considerable variability across models (Fig. 2). While most algorithms demonstrated good results for a large part of the sites compared to an inter-rater DSC in the range of 0.83 averaged over tumor regions[5], reduced segmentation performance and hence a lack of robustness was observed in several sites (including institution IDs 11, 16, 10, and 30), most commonly for the tumor core and enhancing tumor regions. Zooming into the scores for the top-ranked model with ID 15 (Fig. 3) shows that instances of failure were present regardless of whether the respective institution was encountered during training, prompting an investigation into dataset-specific and per-sample factors that impede generalization. This finding is not specific to the model chosen for visualization and a particular tumor region, respectively, as shown in Supplementary Figs. 7 and 12–14.

As a qualitative analysis, we inspected test samples with bad segmentation metrics from the centralized subset and identified the following common, tumor region-specific failure cases:

- Whole tumor (WT): hyperintensities due to other pathologies are labeled as edema (ED) Fig. 4a.
- Enhancing tumor (ET): Small contrast enhancements not directly connected to the largest lesion are missed (Fig. 4b). Moreover, regions are labeled as ET although they are hyperintense both in the T1 and T1-Gd sequences.
- Tumor core (TC): The necrotic/cystic component of the tumor is unclear and seemingly random parts near the ET region are segmented as necrosis (NCR) Fig. 4b, d.

The official FeTS challenge winner was determined among the five original submissions to FeTS 2022. To compare with the previous state-of-the-art brain tumor segmentation algorithms, we included the BraTS 2021 models in a secondary ranking, which resulted in the original FeTS submissions being superseded; the highest three achieved ranks 7 to 9 (Supplementary Table 2). Hence, models submitted to BraTS 2021 maintained their state-of-the-art status, even on the FeTS 2022 test set. Methodological contributions on how to use the provided institution partitioning information during training, which was unavailable for BraTS 2021 models, were not developed by the challenge participants and the submissions differed mostly in network architecture, post-processing, and model ensembling approaches (Table 2). The only algorithm targeting dataset shifts was model 10, which adapts the batch normalization statistics at test time. Consequently, it remains an open question whether information on data shifts during training can enhance algorithmic robustness and adaptability.

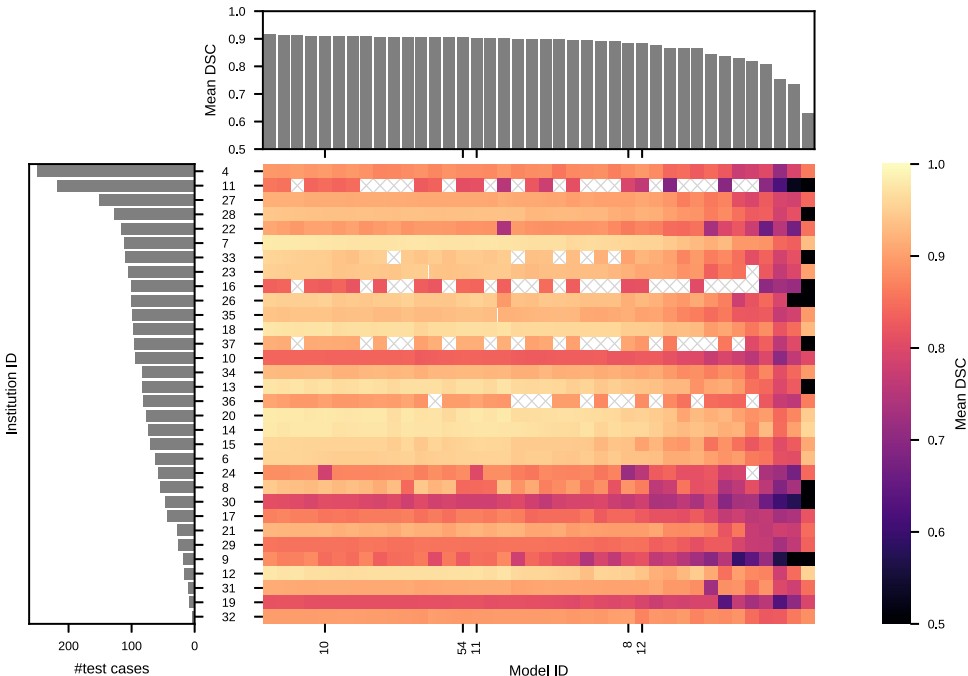

**Fig. 2 | Aggregated results of challenge Task 2 per institution and model.** The figure visualizes test set sizes (left bar plot), mean DSC scores for each institution and submitted model (heatmap; the mean is taken over all test cases and three tumor regions), and mean DSC scores averaged per model (top bar plot). Models are ordered by mean DSC score and official FeTS2022 submissions are marked with ticks. White, crossed out tiles indicate evaluations that could not be completed. The heatmap shows that the performances of the top models are close within each row (i.e., institution) and vary much more between rows. While the drops in mean DSC are moderate, they show that state-of-the-art segmentation algorithms fail to provide the highest segmentation quality for some institutions. Source data are provided as a Source Data file.

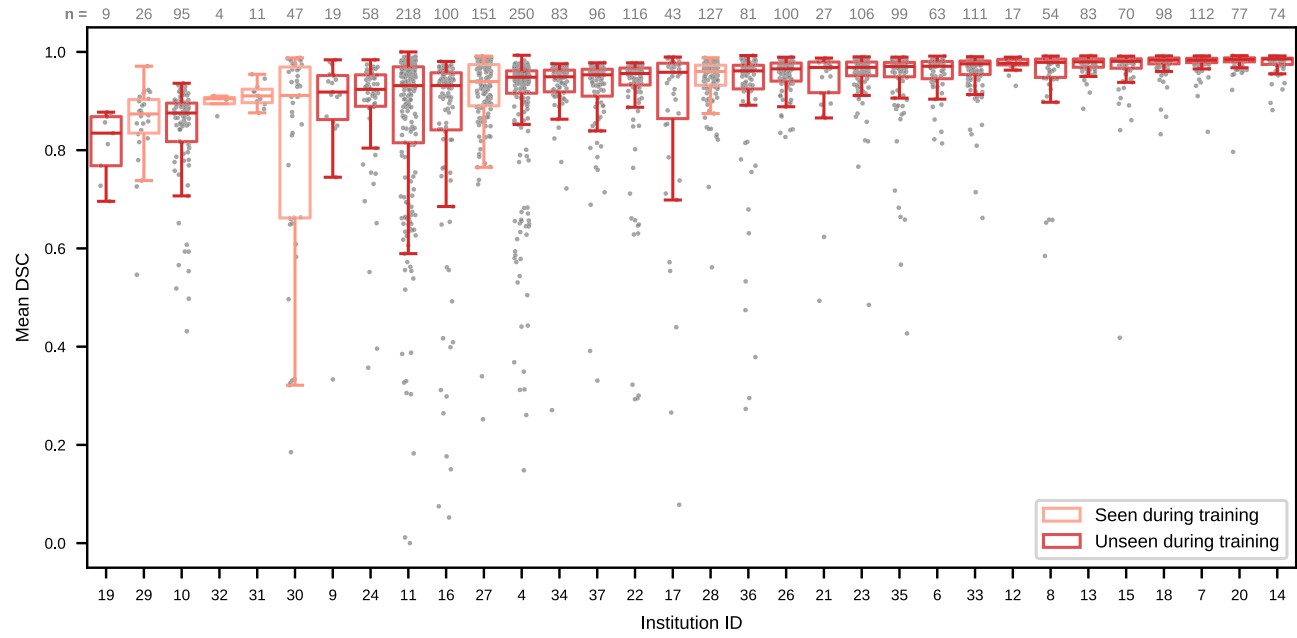

**Fig. 3 | Performance of the top-ranked algorithm for each institution of the test set (Task 2).** Some institutions contributed distinct patients to both the training and testing dataset (marked as seen during training), while others were unseen before testing. Each gray dot represents the mean DSC score over three tumor regions for a single test case, while box plots indicate the median (middle line), 25th, 75th percentile (box) and samples within 1.5 × interquartile range (whiskers) of the distribution. The number of samples $n$ per institution is given above each box. Although median DSC scores are mostly higher than 0.9, institutions with reduced performance or outlier cases exist both within the subset seen during training and the unseen subset. Source data are provided as a Source Data file.

## Heterogeneous systems require pre-determined compatibility solutions

The collaborative, multi-site evaluation process in Task 2 required lots of time and coordination for software setup and resolving technical issues. We initiated the setup process on a small subset of collaborators to test the evaluation pipeline. Common problems encountered during these preliminary tests were collected for later use during the subsequent large-scale setup. After installation, a compatibility test was conducted at each site, evaluating the performance of a reference model on both toy cases and actual local test set data to

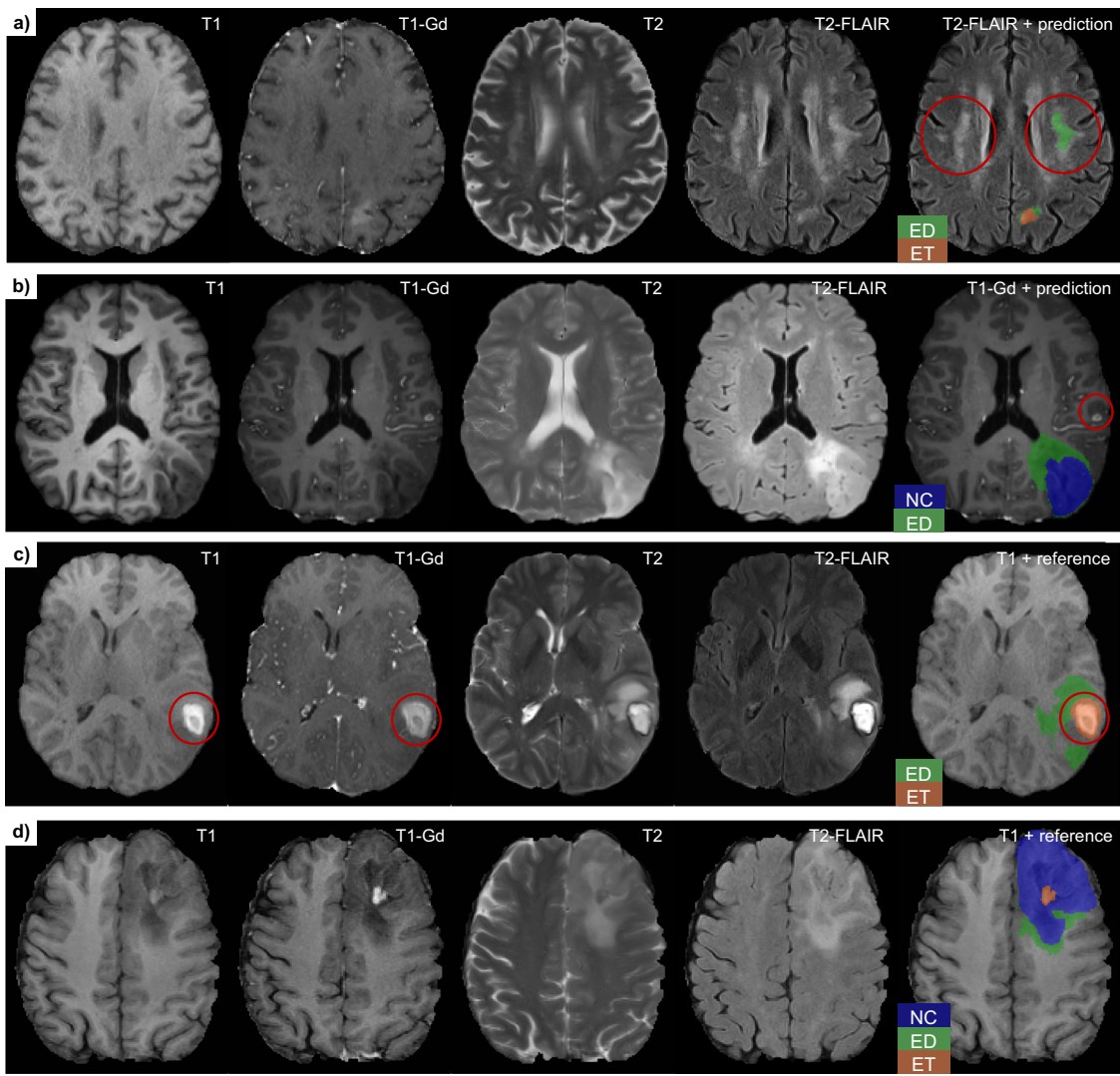

**Fig. 4 | Qualitative examples of common segmentation issues.** Each row shows one case with four MR sequences (T1, T1-Gd, T2, T2-FLAIR) and a segmentation mask overlay in the rightmost column. **a, b** depict errors in the test set prediction of the top-ranked model (ID: 15), while (**c, d**) show training set examples with reference segmentation issues (**c, d**). **a** False positive edema prediction. The hyperintensity is not due to the tumor but a different, symmetric pathology, which is distant from the tumor. **b** A small contrast enhancement is missed by the top-ranked model. It is separate from the larger tumor in the lower right but should be labeled as ET. **c** Since blood products are bright in T1 and T1-Gd, they can be confused with ET. **d** The segmentation of non-enhancing tumor core parts is difficult and often differs between annotators. Label abbreviations: ED edema, NC necrotic tumor core, ET enhancing tumor.

address potential technical or data issues, respectively. Despite these measures, the setup of the evaluation system across all sites spanned several weeks. Numerous and diverse technical issues arose due to the inherent heterogeneity of systems, which were fixed with remote support through the organizers, mostly based on shared log files, emails and video calls. This resulted in slow feedback loops and revealed communication as a primary bottleneck. In contrast, inference time was not a major limitation and could be adapted to the challenge time frame with suitable runtime limits. For example, the total inference time for all 41 models on 100 subjects, using a single-GPU reference hardware, amounted to 86 hours. In conclusion, our experiences underscore the need for extensive technical monitoring and support. The implementation of enhanced error reporting tools holds the potential to accelerate the setup phase by facilitating the fast resolution of errors.

Ensuring compatibility with heterogeneous GPU hardware within the federation emerged as an important consideration during the challenge. To combat this, we recommended a specific base Docker image for official submissions, which was executed successfully across all participating sites. Several data contributors, however, reported issues related to GPU compatibility on converted BraTS submissions, resulting in the missing model evaluations from Fig. 2. This experience highlights the importance of pre-determined compatibility solutions and assessment of the diverse GPU hardware present in the cohort.

### Reference Segmentation is not always the Gold Standard
Annotation quality is crucial in every challenge, but even more difficult to control in a collaborative, multi-site evaluation as in Task 2. To assess this aspect in the FeTS challenge, reference segmentations for test samples that could be shared with the organizers after the challenge (1201 patients from 16 institutions) were screened for major annotation errors through visual inspection by one of the challenge organizers. In total, major annotation errors were detected in 125 cases (10.4%), which were excluded from the final analysis. These were distributed across institutions, with a median of 5 erroneous cases per site.

**Table 2 | Ranking and characteristics of all algorithms evaluated in Task 2**

| Model ID | Rank | Architecture | Loss | Post-processing | Ensembling | nnU-Net |
|---|---|---|---|---|---|---|
| 15 | 1 | U-Net, larger encoder | CE, batch Dice, region-based | ET (small to NCR) | 10 | Yes |
| 35 | 2 | U-Net, larger encoder, multi-scale skip block | Focal loss, Jaccard, region-based | – | 30 | No |
| 37 | 3 | U-Net | CE, Dice, Top-K, region-based | – | 5 | Yes |
| 38 | 4 | U-Net, residual blocks, transformer in bottleneck | CE, Dice | ET (small to NCR) | 3 | Yes + other |
| 16 | 5 | U-Net | CE, Dice | ET (drop disconnected), TC (fill surrounded), WT (drop small components) | 5 | Yes |
| 14 | 6 | U-Net, larger encoder | CE, batch Dice, region-based | ET (small to NCR) | 5 | No |
| 11 | 7 | U-Net | CE, Dice | TC (fill surrounded) | 5 | Yes |
| 54 | 8 | CoTr, HR-Net, U-Net, U-Net++ | CE, Dice, Hausdorff, region-based | ET (small to NCR) | 5 | Yes + other |
| 10 | 9 | U-Net | CE, Dice, region-based | ET (small to NCR) | 5 | Yes |
| 31 | 10 | U-Net, larger encoder, residual blocks | Dice, focal loss | ET (small to NCR) | 5 | No |
| 51 | 11 | HNF-Net | CE, generalized Dice, region-based | ET (small to NCR) | 5 | No |
| 33 | 12 | U-Net, multiple encoders | CE, Dice, region-based | ET (small to NCR) | 4 | No |
| 46 | 13 | U-Net | CE, Dice, generalized Wasserstein Dice | – | 8 | No |
| 40 | 14 | U-Net, larger encoder, residual blocks | Dice, region-based | ET (small to NCR) | 4 | No |
| 27 | 15 | U-Net, modality co-attention, multi-scale skip block, transformer in bottleneck | CE, region-based | ET (drop small components) | – | No |
| 44 | 16 | U-Net | CE, Dice, region-based | ET (convert to NCR based on auxiliary network), drop small components | 10 | Yes + other |
| 19 | 17 | U-Net | CE, Dice, batch Dice, region-based | ET (small to NCR) | 15 | Yes + other |
| 32 | 18 | U-Net | Batch Dice, region-based | ET (small to neighboring label), drop small components | 5 | No |
| 42 | 19 | – | – | – | – | – |
| 18 | 20 | HarDNet | CE, Dice, focal loss, region-based | – | 3 | No |
| 48 | 21 | U-Net, attention | Dice, region-based | – | 1 | No |
| 25 | 22 | U-Net, attention | CE, Dice, region-based | – | 1 | No |
| 13 | 23 | – | – | – | – | – |
| 26 | 24 | U-Net, multiple decoders | CE, Dice, region-based | TC (remove outside of WT), drop small components, morph. closing | 1 | No |
| 30 | 25 | 2-stage, 2D, CNN, U-Net, U-Net++, residual blocks | Dice | – | 29 | No |
| 41 | 26 | CNN, neural architecture search | CE, Dice, region-based | – | 5 | No |
| 8 | 27 | Swin Transformer | CE, Dice, VAT, region-based | – | 1 | No |
| 12 | 28 | U-Net | Dice, region-based | – | 1 | No |
| 47 | 29 | U-Net | CE, Dice | – | 1 | No |
| 22 | 30 | 2D, U-Net, attention, residual blocks | CE, Dice | – | – | No |
| 45 | 31 | 2-stage, U-Net, residual blocks | CE, Dice, region-based | ET (small to NCR) | 5 | No |
| 52 | 32 | U-Net, attention, residual blocks | Dice, region-based | – | 5 | No |
| 36 | 33 | 2D, U-Net, residual encoder | Dice | – | 1 | No |
| 23 | 34 | 2D, U-Net, residual encoder, transformer | CE, Dice, region-based | – | 1 | No |
| 39 | 35 | 2-stage, U-Net | – | – | 1 | No |
| 43 | 36 | U-Net, multi-stage | BCE | fill holes | 1 | No |
| 21 | 37 | 2D, U-Net++ | Dice, boundary distance | – | 3 | No |
| 28 | 38 | 2-stage, CNN, Graph NN | CE | – | 1 | No |
| 53 | 39 | CNN, larger encoder, residual blocks | Dice, boundary, region-based | ET (small to NCR) | 1 | No |
| 29 | 40 | 2D, U-Net | Dice | – | 1 | No |
| 24 | 41 | – | – | – | – | – |

Four institutions were not used for ranking, as many models could not be evaluated on them due to technical problems. Brief explanations of the algorithm characteristics are provided in the participants' methods section. '-' denotes that nothing was reported for this field. *CNN* convolutional neural network, *(B)CE* (binary) cross-entropy, *VAT* virtual adversarial training.

A diversity of errors was observed, including empty or extremely noisy masks, inaccurately hand-drawn masks, duplicate scans, and image errors related to registration or skull-stripping. Two more subtle but common issues were the presence of bright blood products and the extent of the tumor core (TC) region. In some patients, bleeding can occur inside or outside of the tumor. Blood can be recognized as hyper-intensity in T1. It was wrongly labeled as ET in 43 cases, possibly because blood products also appear hyper-intense in the T1-Gd sequence (Fig. 4c). Furthermore, the extent of the TC region as defined in the BraTS annotation protocol compared to the clinical lingo might be considered inherently subjective, because this region may contain non-enhancing tumor parts, which are hard to distinguish from the edematous/infiltrated regions (Fig. 4d). As inter-annotator variations caused by this are consistent with the annotation protocol, we did not consider cases with non-enhancing parts erroneous but note that 46 cases might fall into this category. Both issues above appear also in the training set, which could explain why the results did not change significantly after excluding these cases. Our analysis further highlights the common concern in the domain of medical image segmentation, where the reference segmentations used for algorithmic evaluation are not necessarily what can be considered the ground truth. This is further exacerbated by considering the inter- and intra-rater variability in creating such reference segmentations[5], as well as even taking into consideration the variability in the interpretation of the clinical response assessment for neuro-oncology criteria[51].

## Discussion

In the challenge task on FL (Task 1), the collective insights across participating teams showed that improvements in segmentation performance and training efficiency can coexist by leveraging selective collaborator sampling methods. Trust in these results is further cemented by the reproducible nature of Task 1, which reliably exhibited the same pattern across teams leveraging this type of technique.

The Task 1 submissions also presented a variety of solutions for adaptively aggregating the parameters of locally trained models. Common patterns found in their algorithm characteristics show that methods similar to FedAvg[11] are still the predominant approach for weight aggregation in FL. In 2022, one team deviated from this approach by using an aggregation method motivated by multi-objective minimization theory[39], but reported inferior performance compared to FedAvg. Another alternative approach, in which models transfer and train sequentially from site to site instead of training simultaneously while communicating only with a single trusted global server, was explored in the 2021 instance of the FeTS challenge[47]. The overall performance was consistently lower than the FedAvg-based methods of simultaneous training, meaning the additional communication cost and security risk of every site communicating with every other site is not a warranted alternative.

As a benchmark of FL algorithms in a challenge setting, the FeTS challenge Task 1 also has limitations. The proposed evaluation protocol takes into account the final segmentation performance and the FL efficiency of submitted algorithms through the segmentation metrics and convergence score metric, respectively. The computation of the convergence score was based on simulated federated round times, which depended mostly on the number of data samples at each institution. While the total simulated FL runtime was limited for the FeTS challenge, there may be different limiting factors for other applications, such as constraints on the total communication budget or the communication bandwidth. Future challenges and FL benchmarks should also take these aspects into account in their evaluation strategies, to guarantee a fair and meaningful comparison of FL algorithms.

The challenge design for Task 1 focused on methodology for federated weight aggregation and client selection and did not allow modifying other aspects like the local optimization procedure or the model architecture. These constraints were chosen to foster innovation in these specific parts of the FL algorithm and to make performance gains more attributable. We also wanted to keep the complexity and hence the barrier to participation low. Furthermore, simulating the total FL time becomes increasingly difficult if more degrees of freedom are introduced in the methods. Nevertheless, giving participants more flexibility in their algorithm design is an interesting future direction of FL challenges, as it could shed light on the relative importance of other algorithm components in FL for medical images.

For the collaborative, multi-site validation (Task 2), we formulated two research questions, asking whether brain tumor segmentation is solved in the wild, and what are the pitfalls of competitions using multi-site evaluation. In light of our results, we conclude the following.

The FeTS 2022 dataset possesses even higher diversity than BraTS 2021, marking a significant step towards evaluation in the wild. Existing BraTS models generalized well to unseen sites (in terms of median performance), even though they were not specifically developed for a multicentric deployment. This highlights how a large and diverse training set like BraTS 2021 can be sufficient for good out-of-sample generalization. However, different segmentation performance levels were observed between evaluation sites, and for many of these, individual test cases exhibited failures that were visually confirmed as not related to inter-rater differences. All of this indicates that the robustness and reliability of these models could be further improved.

Our experience during the multi-site evaluation highlights challenges and opportunities for using this collaborative evaluation protocol in biomedical competitions: (i) Extensive communication and coordination are necessary to organize such a competition, making it a substantially time-consuming endeavor. (ii) From the annotation quality results, it is clear that efficient tools for quality control are needed, in particular for challenges with a large set of independent data contributors and annotators. While this study relied on human visual inspection, we also found that the DSC score between the prediction of a state-of-the-art model (i.e., the BraTS 2021 winning solution) and the reference segmentation of the FeTS 2022 test set can help to detect erroneous segmentations: When sorting the test samples by this score, the samples with the lowest 20.0% DSC scores contained 54.4% of the samples with major annotation errors (Supplementary Fig. 18). (iii) The scarcity of meta-data for the test set limited the scope of our analysis. Insights into dataset characteristics and sources of failures observed in multi-site validation studies are only possible with additional test-case-specific information like meta-data or individual images and predictions, which often remain unavailable due to privacy concerns.

To continue the FeTS challenge Task 2 (generalization) in the future, the existing infrastructure can be re-used, decreasing the initial setup effort. However, changes in staff, hardware, or software at individual sites are potential hurdles for maintaining a multi-site benchmark over a long time. Benchmarking initiatives like MedPerf[31] can help in the technical maintenance of challenges with multi-site evaluation. From the 41 evaluated models, only 5 were original submissions to Task 2, from which a single team addressed distribution shifts methodologically. To increase participation and innovation in future competitions, we think it is essential to emphasize the generalization aspects of Task 2 more and to provide researchers with more opportunities to study distribution shifts in the training data. Balancing the training set with respect to the number of cases per institution could be helpful, for example, or additional meta-data on imaging or patient characteristics for each case. Similarly, balanced test data collection is another future direction. Although the FeTS challenge's test set is large, the number of cases varies widely per site and geographical region. Therefore, future efforts should aim to collect more samples for currently under-represented regions or patient populations.

If the aforementioned hurdles associated with collaborative, multi-site validation can be addressed, the reward is a drastic increase in dataset size and diversity, as the distributed setup enables data-sharing from collaborators in a privacy-preserving manner not possible in conventional centralized setups. Multi-site evaluation is therefore well suited for the concept of a phase 2 challenge (competition), which takes place after a phase 1 challenge with a relatively smaller and less diverse dataset has been concluded. Such phase 2 challenges enable the identification of sites among the large federation in which state-of-the-art algorithms show reduced performance and further analysis of where they fail and why.

## Methods

This research complies with all relevant ethical regulations. Informed consent in signed form was obtained from all subjects at the respective institutions that contributed training and validation data, and the protocol for releasing the data was approved by the institutional review board of the data-contributing institution. The provided training and validation data describe mpMRI scans, acquired from: University of Pennsylvania (PA, USA), University of Alabama at Birmingham (AL, USA), Heidelberg University (Germany), University of Bern (Switzerland), University of Debrecen (Hungary), Henry Ford Hospital (MI, USA), University of California (CA, USA), MD Anderson Cancer Center (TX, USA), Emory University (GA, USA), Mayo Clinic (MN, USA), Thomas Jefferson University (PA, USA), Duke University School of Medicine (NC, USA), Saint Joseph Hospital and Medical Center (AZ, USA), Case Western Reserve University (OH, USA), University of North Carolina (NC, USA), Fondazione IRCCS Instituto Neuroligico C. Besta, (Italy), Ivy Glioblastoma Atlas Project, MD Anderson Cancer Center (TX, USA), Washington University in St. Louis (MO, USA), Tata Memorial Center (India), University of Pittsburg Medical Center (PA, USA), University of California San Francisco (CA, USA), Unity Health, University Hospital of Zurich.

This section describes the FeTS Challenge 2022. A description of how the FeTS Challenge 2021 differed from it is provided in the Supplementary Note 5.

### Challenge datasets

**Data sources.** We leverage data from the BraTS challenge[4,5,52–54], and from 32 collaborators of the largest to-date real-world federation[28]. The following sections apply to both of them unless otherwise noted. Both sources contain mpMRI scans routinely acquired during standard clinical practice along with their reference annotations for the evaluated tumor sub-regions. These are augmented with meta-data of the scans' partitioning in an anonymized manner. Each case describes four structural mpMRI scans for a single patient at the pre-operative baseline time point. The exact mpMRI sequences included for each case are (i) native T1-weighted (T1), (ii) contrast-enhanced T1 (T1-Gd), (iii) T2-weighted (T2), and (iv) T2 Fluid Attenuated Inversion Recovery (T2-FLAIR).

**Data preprocessing.** The preprocessing pipeline from the BraTS challenge is applied in the FeTS challenge, too. Specifically, all input scans (i.e., T1, T1-Gd, T2, T2-FLAIR) are rigidly registered to the same anatomical atlas (i.e., SRI-24[55]) using the Greedy diffeomorphic registration algorithm[56], ensuring a common spatial resolution of 1 mm³. After registration, brain extraction is done to remove any apparent non-brain tissue, using a deep learning approach specifically designed for brain MRI scans with apparent diffuse glioma[57]. All preprocessing routines have been made publicly available through the Cancer Imaging Phenomics Toolkit (CaPTk)[58–60] and the FeTS tool[61].

**Annotation protocol.** The skull-stripped scans are used for annotating the brain tumor sub-regions. The annotation process follows a pre-defined clinically approved annotation protocol[3,4], which was provided to all clinical annotators, describing in detail the radiologic appearance of each tumor sub-region according to the specific provided MRI sequences. The annotators were given the flexibility to use their tool of preference for making the annotations, and also follow either a complete manual annotation approach or a hybrid approach where an automated approach is used to produce some initial annotations followed by their manual refinements. The summarized definitions of the tumor sub-regions communicated to annotators are:

1. The enhancing tumor (ET) delineates the hyperintense signal of the T1-Gd sequence compared to T1, after excluding the vessels.
2. The tumor core (TC) represents what is typically resected during a surgical operation and includes ET as well as the necrotic tumor core (NCR). It outlines regions appearing dark in both T1 and T1-Gd images (denoting necrosis/cysts) and dark regions in T1-Gd and bright in T1.
3. The farthest tumor extent, also called whole tumor (WT), consists of the TC as well as the peritumoral edematous and infiltrated tissue (ED). WT delineates the regions characterized by the hyperintense abnormal signal envelope on the T2-FLAIR sequence.

The provided segmentation labels have values of 1 for NCR, 2 for ED, 4 for ET, and 0 for everything else.

For the BraTS data, each case was assigned to a pair of annotator-approver. Annotators spanned across various experience levels and clinical/academic ranks, while the approvers were the 2 experienced board-certified neuroradiologists (with more than 13 years of experience with glioma). Annotations produced by the annotators were passed to the corresponding approver, who was then responsible for signing off these annotations. Specifically, the approver would review the tumor annotations in tandem with the corresponding mpMRI scans, and send them back to the annotators for further refinements if necessary. This iterative approach was followed for all cases until their annotations reached satisfactory quality (according to the approver) for being publicly available and noted as final reference segmentation labels for these scans.

Collaborators from the FeTS federation were asked to use a semi-automatic annotation approach, leveraging the predictions of an ensemble of state-of-the-art BraTS models. Specifically, collaborators were supplied with the FeTS tool[61], containing pre-trained models of the DeepMedic[62], nnU-Net[63], and DeepScan[64] approaches trained on the BraTS data, with label fusion performed using the Simultaneous Truth and Performance Level Estimation (STAPLE) algorithm[65,66]. Refinements of the fused labels were then performed by neuroradiology experts at each site according to the BraTS annotation protocol[4]. Sanity checks to ensure the integrity and quality of the annotations were performed in a preceding FL study[28].

**Training, validation, and test case characteristics.** Training and Validation sets for the FeTS challenge were gathered from the BraTS dataset, sampling a specific subset of radiographically appearing glioblastoma while excluding cases without an apparent enhancement. The exact numbers can be found in Table 3. Training cases encompass the mpMRI volumes, the corresponding tumor sub-region annotations, as well as a pseudo-identifier of the site where the scans were acquired. In contrast, validation cases only contain the unannotated mpMRI volumes. We provided two schemas to the participants for partitioning the provided data and used a third partitioning internally for re-training submissions before the test set evaluation (details in Supplementary Fig. 1):

1. Geographical partitioning by institution (partitioning 1, 23 sites)
2. Artificial partitioning using imaging information (partitioning 2, 33 sites), by further sub-dividing each of the 5 largest institutions in partition 1 into three equally large parts after sorting samples by their whole tumor size.

**Table 3 | Overview of the number of cases and institutions in the training, validation, and test sets**

|  | Training | Validation | Test (Task 1) | Test (Task 2) |
|---|---|---|---|---|
| Source | BraTS21 | BraTS21 | BraTS21 | BraTS21 + FeTS |
| No. cases | 1251 | 219 | 570 | 2625 |
| No. sites | 23[a] | n/a | n/a | 32 |
| Access | Public (img, seg) | Public (img) | Organizers | Data owners |

The centralized, multi-centric data from the Brain Tumor Segmentation Challenge 2021 (BraTS21)[3] is used for benchmarking FL methods (Task 1). Additionally, for Task 2 the testing data is augmented with distributed data from the FeTS initiative[28], increasing size and geographical diversity drastically. *img* imaging data, *seg* reference segmentations.
[a]based on partitioning 1.

3. Refined geographical partitioning (partitioning 3, 29 sites), which was generated as a refinement of the geographical partitioning (partitioning 1), by subdividing the largest institution into seven parts. This institution comprises a system of hospitals in close geographical proximity, which were combined for partitioning 1. For partitioning 3, they were re-grouped into seven pseudo-institutions.

Testing datasets were also gathered from BraTS and the FeTS federation collaborators but were not shared with the challenge participants. Access to the centralized test datasets was exclusive to Task 1 organizers, while the datasets for Task 2 remained decentralized throughout the competition, inaccessible for the Task 2 organizer. This collaborative, multi-site evaluation approach scaled up the size and diversity of the test dataset compared to the BraTS 2021 challenge significantly (Supplementary Fig. 11).

## Performance evaluation

Predictions of the submitted segmentation algorithms were required to follow the format of the provided reference segmentations. Segmentation quality is assessed on the ET, TC, and WT sub-regions, corresponding to the union of labels {4}, {1, 4}, and {1, 2, 4}, respectively. For each region, the predicted segmentation is compared with the reference segmentation using the following metrics:

- Dice similarity coefficient (DSC), which measures the extent of spatial overlap between the predicted masks ($\hat{Y}$) and the provided reference ($Y$), defined by

$$\text{DSC} = \frac{2|Y \cap \hat{Y}|}{|Y| + |\hat{Y}|}. \tag{1}$$

DSC scores range from 0 (worst) to 1 (best). The DSCs of the three individual tumor regions can be averaged to obtain a mean DSC.

- Hausdorff distance (HD), which quantifies the distance between the boundaries of the reference labels against the predicted label. This makes the HD sensitive to local differences, as opposed to the DSC, which represents a global measure of overlap. For brain tumor segmentation, local differences may be crucial for properly assessing segmentation quality. In this challenge, the 95th percentile of the HD between the contours of the two segmentation masks is calculated, which is more robust to outlier pixels:

$$\text{HD}_{95}(\hat{Y}, Y) = \max \left\{ \begin{array}{cc} P_{95\%} \, d(\hat{y}, Y), & P_{95\%} \, d(y, \hat{Y}) \\ \hat{y} \in \hat{Y} & y \in Y \end{array} \right\}, \tag{2}$$

where $d(a, B) = \min_{b \in B} ||a - b||$ is the distance of $a$ to set $B$. Lower distances correspond to more accurate boundary delineations.

- Convergence Score is an additional metric used for Task 1 only. It measures how quickly algorithms are able to reach a desired segmentation performance. Methods with fast convergence allow to stop training earlier, thus saving communication and computation resources and enhancing the efficiency of federated training. To calculate the convergence score, in each round of an FL experiment, the mean DSC on a fixed validation split (20%) of the official training data and the simulated round time $T$ are computed. Details on how $T$ is simulated are in the FL framework methods. Over the course of an experiment, this results in a DSC-over-time curve. The validation DSC can in some cases decrease at later times (e.g., due to overfitting or randomness in the optimization), but as the model with the best DSC is used as the final model, such a decrease should not be penalized. Therefore, a projected DSC curve is computed as $\text{DSC}_{\text{proj}}(t) = \max_{t' \leq t} \text{DSC}(t')$. The final convergence score metric is calculated as the area under that projected DSC-over-time curve. Higher values of this metric indicate enhanced convergence and, thus, the best FL approach. To standardize the time-axis for the convergence score among the Task 1 participants, all FL experiments performed during the challenge were limited to one week of simulated total time, which was a realistically feasible duration based on the experience from the FeTS initiative[28]. The FL runs were terminated once the simulated time exceeded one week and the model with the highest validation score before the last round was used as the final model, to make sure that a long last round exceeding the time limit does not benefit the participant.

## Task 1: federated training (FL weight aggregation methods)

**Model architecture.** To focus on the development of aggregation methods, we needed a pre-established segmentation model architecture. Based on current literature indications, we picked U-Net[67] with residual connections, which has shown robust performance across multiple medical imaging datasets[57,63,68–71]. The U-Net architecture consists of an encoder, comprising convolutional layers and down-sampling layers (applying max-pooling operation), and a decoder of upsampling layers (applying transpose convolution layers). The encoder-decoder structure contributes in capturing information at multiple scales/resolutions. The U-Net also includes skip connections, which consist of concatenated feature maps paired across the encoder and the decoder layers, to improve context and feature re-usability, boosting overall performance.

**Federated learning framework.** We employ the typical aggregation server FL workflow[14], in which a central server (aggregator) exchanges model weights with participating sites (collaborators), which are simulated for the FeTS challenge Task 1 on a single machine using the real-world multicentric data described in the challenge datasets methods. This process is repeated in multiple FL-based training rounds. At the start of a single round, each collaborator locally validates the model received from the aggregator. Each collaborator then trains this model on their local data to update the model gradients. The local validation results along with the model updates of each site are then sent to the aggregator, which combines all model updates to produce a new consensus model. This model is then passed back to each collaborator and a new federated round begins. Following extensive prior literature[33,63,71,72], the final model for each local institutional training is chosen based on the best local validation score at pre-determined training intervals, i.e., rounds.

To guarantee fair competition, all challenge participants were required to use an implementation of this FL framework based on

PyTorch and openFL[73,74] provided by the organizers. Modifications were allowed in the following components:

- Aggregation method: Participants could customize how weights from the current training round are combined into a consensus model.
- Collaborator selection: Instead of involving all collaborators in each round, participants can selectively sample collaborators, for example based on validation metrics or round completion time.
- Hyperparameters for local training: In each FL round, participants could adjust the values of two essential FL parameters, the learning rate of the stochastic gradient descent (SGD) optimizer, and the number of epochs per round.

Efficiency is an important practical aspect of FL with its inherent communication and computation constraints. As described in the evaluation section, we take this into account in the FL benchmarking framework by limiting wall clock runtime and by evaluating the convergence score metric, both of which require the realistic simulation of FL round durations. To make this simulation as realistic as possible, we used a subset of the real-world times measured in the FeTS initiative[28]. Note that the simulated time is different from the program runtime; it is rather an estimate of the wall time such an FL experiment would take in a real federation similar to the FeTS initiative. Specifically, we subdivide simulated time into: training time $T_{\text{train}}$, validation time $T_{\text{val}}$, model weight download $T_{\text{down}}$ and upload time $T_{\text{up}}$. In each round, the simulated time for each collaborator $k$ is

$$T_k = T_{\text{down}, k} + T_{\text{up}, k} + T_{\text{val}, k} \cdot N_{\text{val}, k} + T_{\text{train}, k} \cdot N_{\text{train}, k} \quad (3)$$

and the total time for each round is $\max_k \{T_k\}$. To simulate a realistic FL setup, $T_{x,k}$ was sampled from a normal distribution: $T_{x,k} \sim \mathcal{N}(\mu_{x,k}, \sigma_{x,k})$, where x can be replaced with train/val/down/up. The parameters of the normal distribution are fixed but different for each client $k$, and based on time measurements in a previous real-world FL study, which used the same model[28]. Random seeds guarantee that these are identical for all FL experiments, so that all participants use the same timings.

**Ranking.** Before evaluating the submissions on the Task 1 test set, all algorithms were re-trained by the organizers, to ensure reproducible results and to prevent data leakage between federated sites. As the participants should develop generalizable FL algorithms that do not overfit on a particular collaborator, the unseen, refined geographical partitioning (partitioning 3) was used. Then, based on the measured metric values, a ranking methodology akin to the BraTS challenges was employed. All teams are ranked for each of the $N$ test cases, 3 tumor regions, and 2 segmentation metrics separately, yielding $N \cdot 3 \cdot 2$ rankings. Additionally, the teams' performance was evaluated based on the convergence score, which was incorporated into each case-based ranking with a factor of 3, due to the importance of efficiency in FL. This results in a total of $N \cdot 3 \cdot 3$ ranks summed per team. The final ranking was determined by summing all individual rankings per team.

### Task 2: multi-site evaluation of generalization in the wild
**Organization.** In the training phase, the participants were provided the training set including information on the data origin. They could explore the effects of data partitioning and distribution shifts between contributing sites, to develop tumor segmentation algorithms that generalize to institutional data not present in the training set. Note that training on pooled data was allowed in Task 2, enabling the development of methods that optimally exploit meta-information of data origin.

In the validation phase, participants could evaluate their model on the validation set to estimate in-distribution generalization. For domain generalization there may be better model selection strategies than an in-distribution validation set[75], which opened up further research opportunities for the participants.

Participants could submit their inference code as Docker containers[76] to the Synapse challenge website at https://www.synapse.org/fets. The latest submission before the deadline was chosen as the final submission. All submissions were tested in an isolated environment on cloud computing infrastructure at DKFZ, which ensures a secure and compliant processing framework and safeguards the host infrastructure from potential malicious attacks. This included the following steps:

1. Convert Docker submissions to singularity container[77], as Docker was not allowed on some of the evaluation sites' IT departments.
2. Run a compatibility testing pipeline, which evaluates the container on a small training subset, using the same software as during the testing phase (described below).
3. Monitor the GPU memory consumption and inference time, which were limited to ensure functionality in the federation.
4. Update the challenge website with the results of the test run and, if successful, upload the container to cloud storage.

Step 2 could also be executed by the participants locally to debug their submission.

In the testing phase, the MedPerf tool[31] was used to evaluate all valid submissions on datasets from the FeTS federation, such that the test data are always retained within their owners' servers.

**Assessment methods (Ranking).** The accuracy of the predicted tumor segmentations is measured with DSC and HD$_{95}$ (Eqs. (1) and (2)). To assess the robustness of segmentation algorithms to cross-institution shifts, we evaluate algorithms per testing institution first and rank them according to their per-institution performances. Specifically, on institution $k$ of $K$, algorithms are ranked in the first step on all $N_k$ test cases, three regions, and two metrics, yielding $N_k \cdot 3 \cdot 2$ ranks for each algorithm. The average over test cases is then used to produce per-institution ranks for each algorithm (rank-then-aggregate approach) and region-metric combination. The final rank of an algorithm is computed from the average of its $K \cdot 3 \cdot 2$ per-institution ranks. Ties are resolved by assigning the minimum rank. This scheme was chosen as it is similar to the BraTS ranking method[4]. Moreover, our ranking method weights each testing institution equally, as they represent distinct dataset characteristics and we want to avoid a strong bias of the ranking to sites with many test cases.

### Description of participants' methods
As described in the results, for task 1 most participants chose a multi-step approach, which computes several independent, normalized weighting terms $p_i$ (step 1) and combines them into an overall weight $\bar{p}$ (step 2). The latter was done either by additive or multiplicative averaging, defined as

$$\bar{p}_{\text{add}}^k = \sum_i \beta_i p_i^k \quad \text{or} \quad \bar{p}_{\text{mul}}^k = \prod_i p_i^k \quad (4)$$

where $p_i^k$ is the weighting term for collaborator $k$ and $\beta_i$ are averaging weights (hyperparameters). The $\bar{p}^k$ are then normalized and used to aggregate local model parameters $w_t^k$ across $K$ collaborators into a global model $w_t^g$ for each FL round $t$:

$$w_{t+1}^g = \sum_{k=1}^{K} \bar{p}^k w_t^k \quad (5)$$

The weighting term that all participants incorporated in their solution was proposed by McMahan et al.[11]: $p_{\text{FedAvg}}^k = N_k / \sum_k N_k$, where $N_k$ is the number of local samples. Most teams introduced additional adaptive

aggregation methods, which change the weighting $p^k(t)$ over the course of federated training rounds $t$.

A summarizing description of the methods contributed by the participating teams is provided below, ordered alphabetically by team name. For Task 2, only the five official submissions are included here. Key components in which the algorithms differ are also presented in Table 1 for Task 1 and Table 2 for Task 2. The algorithm characteristics for Task 2 that stood out in the participants' method descriptions were the network architecture, the loss function, post-processing steps applied to the model's predicted segmentation mask, the number of models used in the final ensemble (ensemble size) and whether they used the nnU-Net framework for their implementation. A complete list of members for each team is given in the Supplementary Note 4.

**Team Flair[39]—Task 1.** This team presented additional dataset splits of varying sizes for prototyping and tested how a federated version of the multiple gradient descent algorithm, which formulates FL as multi-objective optimization[78], performs on the problem. This weight aggregation method ensures that gradient steps are taken only in a direction that does not harm the model performance on individual clients, while also not deviating from the FedAvg weights by more than a hyperparameter $\epsilon$. Full client participation was used in all rounds.

**Team FLSTAR[36]—Task 1.** This team tested how various aggregation strategies improve the learning performance in the context of the non-IID and imbalanced data distribution of the FeTS challenge data (partitioning 2). Their final model used a (normalized) multiplicative average of FedAvg weights $p^k_{FedAvg}$ and local validation loss for aggregating the clients' parameters: $p^k_{L_{val}}(t) = \frac{1}{Z}L(w^k_t)$, where $L(w^k_t)$ is the validation loss after local training and $Z$ a normalization factor. This term can be interpreted as measuring the potential for local optimization, as clients with high loss can still improve more than low-loss clients. For client selection, only the 6 largest sites from partitioning 2 were used, as they were less prone to overfitting.

**Team Gauravsingh[41]—Task 1.** This team implemented an aggregation method inspired by Mächler et al.[44], which uses an arithmetic mean of two (normalized) terms for each client weighting factor: (1) local dataset size as in FedAvg, (2) ratio of local validation loss (here negative DSC) after and before local training $p^k_{CostWAvg}(t) = Z^{-1} \cdot DSC(w^k_t)/DSC(w^g_t)$, where $Z$ normalizes across clients. For client selection, they randomly subdivided all clients into groups of 6 clients and iterated through the groups in each federated round, so that 6 clients are used per round. Every four rounds, the clients were re-grouped.

**Team Graylight Imaging[79]—Task 2.** This team built upon the 3D nnU-Net framework, incorporating a customized post-processing step specifically designed for the TC region. The post-processing method, denoted as FillTC, involves relabeling voxels surrounded by TC to NCR. This iterative post-processing is sequentially applied to each 2D slice, first in the axial direction and subsequently in the coronal and sagittal directions. The rationale behind this approach is grounded in clinical expertise, suggesting that significant tumors typically lack voids of healthy tissue. Furthermore, if a given region is surrounded by NCR or ET, it is deemed to be part of the TC.

**Team HPCASUSC[80]—Task 2.** This team built their model upon a 3D U-Net and added improvements inspired by the BraTS nnU-Net (2020) paper[63]. They used region-based training, which uses the WT, TC, and ET regions as labels during training instead of NCR, ED, and ET. Further, they increased the batch size to 24 and used batch normalization layers instead of instance normalization. Data augmentation consisted of random mirroring, rotation, intensity shift, and cropping.

**Team HT-TUAS[40]—Task 1.** This team introduced a cost-efficient method for regularized weight aggregation, building upon their previous year's submission[42]. For parameter aggregation, the average of FedAvg weighting and a parameter-distance (similarity) weighting was used. Similarity with the average model parameters $\bar{w}_t = \frac{1}{K}\sum_k w^k_t$ is measured with the absolute difference between individual local parameters and average parameter tensors $p^k_{sim}(t) = \frac{1}{Z}|\bar{w}_t - w^k_t|^{-1}$, where the absolute value is applied element-wise. Additionally, the team scaled the individual client weights with a regularization term that is proportional to the parameter difference between the current and previous round. For client selection, they randomly sampled 4 sites per round without replacement and restarted the sampling once all clients participated.

**Team NG research[81]—Task 2.** This resubmission from the BraTS 2021 challenge, makes heavy use of model ensembling. The ensemble comprises five models of diverse architectures, both convolutional and transformer-based, which are combined with mean softmax. Their models were refined by several strategies: Randomized data augmentations, incorporating affine transforms, mirroring, and contrast adjustment, were employed during training to enhance model robustness. Furthermore, a post-processing step was integrated, selectively discarding ET predictions falling below a specified volume threshold, similar to Isensee et al.[63].

**Team rigg[35]—Task 1.** This team developed FedPIDAvg, an aggregation method that is inspired by a proportional-integral-derivative controller. Compared to the predecessor method[44], it adds the missing integral term. The aggregation weight for each client is hence the weighted sum of three terms, normalized with factors $Z$ as necessary: (1) local dataset size identical to FedAvg $p^k_P = p^k_{FedAvg}$, (2) cost reduction (or local improvement), i.e., the difference between local loss of the previous and current round, $p^k_D(t) = \frac{1}{Z_D}(L(w^k_{t-1}) - L(w^k_t))$, (3) sum of the local loss over the past 5 rounds $p^k_I(t) = \frac{1}{Z_I}\sum_{i=1}^{5}L(w^k_{t-i})$, which indicates how much room for improvement remains. Selective sampling was also incorporated, by modeling the sample distribution across clients with a Poisson distribution and randomly dropping outliers, i.e., large clients.

**Team RoFL[37]—Task 1.** This team focused on tackling data heterogeneity among collaborators and the communication cost of training, exploring a combination of server-side adaptive optimization and judicious parameter aggregation schemes. Server optimizers[50] rewrite the model aggregation equation Eq. (5) in the form of a stochastic gradient descent (SGD) update: $w^g_{t+1} = w^g_t - \lambda_s\Delta_t$, where $\lambda_s$ is a server learning rate and $\Delta_t$ the aggregated model update. SGD can then also be replaced with other optimizers. Team RoFL's final submission uses Adam[82] as the server optimizer and takes a two-phase approach: in the first phase, aggregation in $\Delta_t$ is performed with FedAvg. In the second phase, the client learning rate is decreased while the server learning rate $\lambda_s$ increased. Furhermore, the model updates are aggregated with a multiplicative combination of FedAvg weights and a term computed per scalar parameter that is proportional to the inverse absolute difference between local and average model parameter, as in ref. 42. Full client participation was used in all FL rounds.

**Team Sanctuary[38]—Task 1&2.** The solution for Task 1 incorporates three key components. Firstly, model updates are aggregated through inverse distance weighting[83], where the inverse L1 distance between the current and the average model parameters is employed to weight the updates contributed by each site. $p^k_{dist}(t) = \frac{1}{Z}\|\bar{w}_t - w^k_t\|^{-1}$ Here, $\bar{w}_t = \frac{1}{K}\sum_k w^k_t$ is the uniformly averaged model and $Z$ normalizes across collaborators. This aggregation weight is computed for each tensor in

the model and combined with the FedAvg weight and a weight inversely proportional to the local training DSC, which penalizes overfitting clients and lifts the weight of clients with potential for local optimization. To mitigate the impact of slow clients on training efficiency, a client pruning strategy is implemented. In even FL rounds, full client participation is used. In odd FL rounds, the simulated round time of each client from the previous is used to select a subset of clients, by dropping clients that exceed a time threshold, which is set to $0.75 \cdot \bar{t}$, where $\bar{t}$ is the average round time. Additionally, the team adopted a polynomial learning rate schedule to enhance training convergence.

For Task 2, they based their submission on the nnU-Net contribution for BraTS 2020[63], extending it with test-time adaptation through batch normalization (BN) statistics. Unlike the conventional approach of collecting and freezing BN statistics during training, their method leverages test data information to dynamically correct internal activation distributions, particularly addressing domain shift issues. In their approach, test-time BN recalculates BN statistics (mean $\mu$ and standard deviation $\sigma$ per filter map) based on the batch at prediction time. As the algorithm utilized a batch size of 1 during testing, it is similar to instance norm at test time. Furthermore, the team employed an ensemble strategy involving six models trained on distinct training data folds. Each of these models underwent adaptation using test-time BN.

**Team vizviva[84]—Task 2**. This team employed an encoder-decoder architecture based on volumetric vision transformers. In this setup, the encoder partitions a 3D scan into patches, subsequently processing them through layers that amalgamate the outputs of 3D Swin transformer and 3D CSwin transformer blocks[85,86]. For the decoder, 3D Swin transformer blocks and patch expansion layers are utilized to reconstruct the processed information. The training strategy involves a combination of cross-entropy and Dice loss. Additionally, to bolster the model's resilience against adversarial examples, virtual adversarial training introduces an extra loss term.

**Additional information on Task 2 algorithms**. In the FeTS challenge 2022, Task 2, not only official challenge submissions were evaluated, but also 36 models submitted originally to the BraTS challenge 2021[3]. These models are the subset of BraTS 2021 submission that could be converted semi-automatically to the container format used in the FeTS Challenge 2022. Since all of these were described in scientific publications previously, we provide the references to the papers instead of describing each method here in detail in Supplementary Table 4. In the following, Table 2 is supplemented with references and short descriptions of the Task 2 algorithm characteristics:

**Architecture**. The most common backbone used by the submissions was U-Net[67]. Several variations to the basic U-Net were introduced by the teams: Some used larger encoders, with more filters per convolution or more convolutional blocks per stage. Adding residual connections to convolutional blocks[69] was also common. Several algorithms extended the U-Net with different kinds of attention modules. Examples include inserting a transformer in the bottleneck of the U-Net or re-weighting feature maps with attention restricted to the channel/spatial dimensions. Some participants used other CNNs than U-Net, for instance HR-Net[87], HNF-Net[88], U-Net++[89], and HarDNet[90]. Recent hybrid CNN/transformer networks like CoTr[91], Swin transformer[85] were incorporated in some submissions. Finally, a few teams utilized skip connection blocks that combined features from multiple stages or explored splitting the segmentation task into two stages, first segmenting a coarse whole tumor region and then refining the segmentation of this cropped region.

**Loss**. The most common loss functions were Dice (computed either per sample or per batch) and cross-entropy. Similar to the Dice loss,

some teams optimized differentiable versions of segmentation metrics (Jaccard index, generalized Dice, boundary distance, and the generalized Wasserstein Dice loss[92]). Two less common loss functions were TopK loss, which considers only the K pixels with the highest loss, and the focal loss, which down-weights the loss for pixels that are classified correctly with high softmax scores. Finally, one team used virtual adversarial training[93] as an auxiliary, regularizing loss term. Most losses can be calculated either region-based (for each of WT, TC, ET) or for the exclusive labels (ED, NCR, ET).

**Post-processing**. Techniques that refine a model's segmentation output based on prior knowledge specific to the three brain tumor regions were popular in the challenge. Dropping small connected components from the final mask (or replacing them with neighboring predictions) can help to reduce false positives. Morphological operations like closing or hole filling were also applied by some teams. Since TC usually is a compact core within WT, post-processing methods enforced this property, by removing TC parts that extend beyond WT or filling holes inside TC. Finally, potential confusion between ET and NCR was counteracted by converting ET output regions to NCR if they are very small (or for one team, if an auxiliary network suggests this).

## Reporting summary
Further information on research design is available in the Nature Portfolio Reporting Summary linked to this article.

## Data availability
The training and validation data of the FeTS challenge have been deposited in the Synapse platform under accession code syn29264504 [https://www.synapse.org/Synapse:syn29264504] (registration required for download) and, as they are identical to the BraTS 2021 data, are also available via TCIA under DOI 10.7937/jc8x-9874 [https://www.cancerimagingarchive.net/analysis-result/rsna-asnr-miccai-brats-2021/] (free access). The reference segmentations for the validation data as well as the centralized testing data for the challenge are protected and are not available because they will be re-used in future competitions, which are only fair if evaluation sets are not public. Furthermore, decentralized testing data from the federated institutions are protected and are not available due to data sharing restrictions of the individual institutions. The challenge results data generated in this study are published as a source data file. The source data file contains raw data underlying each figure, two example training cases, and the full challenge metric results for both tasks. Source data are provided with this paper.

## Code availability
To enable reproducibility, all tools, pipelines, and methods have been released through the Cancer Imaging Phenomics Toolkit (CaPTk)[58–60], MedPerf (https://github.com/mlcommons/medperf/tree/fets-challenge)[31] and the FeTS tool (https://github.com/FETS-AI/Front-End/). Challenge-specific instructions are available on the challenge website (https://www.synapse.org/fets). Challenge-specific code for developing and testing algorithms, creating the analysis figures in the article and computing the rankings are publicly available (https://github.com/FETS-AI/Challenge)[94]. That repository consists of components with different licenses, ranging from BSD-style to Apache-2, all approved by the open-source initiative.

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

## Acknowledgements

We would like to thank Manuel Wiesenfarth and Paul F. Jäger (DKFZ) for helpful discussions. Research reported in this publication was partly funded by the Helmholtz Association (HA) within the project "Trustworthy Federated Data Analytics" (TFDA) (funding number ZT-I-OO1 4), and partly by the National Institutes of Health (NIH), under award numbers NCI:U01CA242871 (PI: S.Bakas) and NCI:U24CA279629 (PI: S.Bakas). K. Kushibar holds the Juan de la Cierva fellowship with a reference number FJC2021-047659-I. This work was supported in part by Hong Kong Research Grants Council Project No. T45- 401/22-N. Team HT-TUAS was partly funded by Business Finland under Grant 33961/31/ 2020. They also acknowledges the CSC-Puhti super-computer for their support and computational resources during FeTS 2021 and 2022. N. D. Forkert was supported by the Canadian Institutes of Health Research (CIHR Project Grant 462169). Jakub Nalepa was supported by the Silesian University of Technology funds through the Excellence Initiative–Research University program (Grant 02/080/SDU/10-21-01), and by the Silesian University of Technology funds through the grant for maintaining and developing research potential. Research reported in this publication was partly funded by R21EB030209, NIH/NIBIB (PI: Y. Yuan), UL1TR001433, NIH/NCATS, a research grant from Varian Medical Systems (Palo Alto, CA, USA) (PI: Y. Yuan). Y. Yuan also acknowledges the generous support of Herbert and Florence Irving/the Irving Trust. Z. Jiang was supported by National Cancer Institute (UG3 CA236536). H. Mohy-ud-Din was supported by a grant from the Higher Education Commission of Pakistan as part of the National Center for Big Data and Cloud Computing and the Clinical and Translational Imaging Lab at LUMS. M. Kozubek was supported by the Ministry of Health of the Czech Republic (grant NU21-08-00359 and conceptual development of research organization FNBr-65269705) and Ministry of Education, Youth and Sports of the Czech Republic (Project LM2023050). Václav Vybíhal was supported by MH CZ - DRO (FNBr, 65269705). Y. Gusev was supported by CCSG Grant number: NCI P30 CA51008. P. Vollmuth was supported by Deutsche Forschungsgemeinschaft (DFG, German Research Foundation) - Project-ID 404521405, SFB 1389 - UNITE Glioblastoma, Work Package C02, and Priority Programme 2177 "Radiomics: Next Generation of Biomedical Imaging" (KI 2410/1-1 | MA 6340/18-1). B. Landman was supported by NSF 2040462. A. Rao was supported by the NIH (R37CA214955-01A1). A. Falcão was supported by CNPq 304711/ 2023-3. P. Guevara was supported by the ANID-Basal proyects AFB240002 (AC3E) and FB210017 (CENIA). Research reported in this publication was partly funded by the NSF Convergence Accelerator - Track D: ImagiQ: Asynchronous and Decentralized Federated Learning for Medical Imaging, Grant Number: 2040532, and R21CA270742 (Period of Funding: 09/15/20 - 05/31/21). Martin Vallières acknowledges funding from the Canada CIFAR AI Chairs Program. Stuart Currie receives salary support from a Leeds Hospitals Charity (9R01/1403) and Cancer Research UK (C19942/A28832) grants. Kavi Fatania is a 4ward North Clinical PhD fellow funded by Wellcome award (203914/Z/16/Z). Russell Frood is a Clinical Trials Fellow funded by CRUK (RCCCTF-Oct22/ 100002). This work was funded in part by National Institutes of Health R01CA233888 and the grant NCI:U24CA248265. The content of this publication is solely the responsibility of the authors and does not represent the official views of the HA, or the NIH. U.Baid, S.Pati, and S.Bakas conducted part of the work reported in this manuscript at their current affiliations, as well as while they were affiliated with the Center for Artificial Intelligence and Data Science for Integrated Diagnostics (AI2D) and the Center for Biomedical Image Computing and Analytics (CBICA) at the University of Pennsylvania, Philadelphia, PA, USA.

## Author contributions

Study conception: M. Zenk, U. Baid, S. Pati, K. Maier-Hein, S. Bakas Development of software used in the study: M. Zenk, S. Pati, B. Edwards, M. Sheller, P. Foley, A. Aristizabal, A. Gruzdev, S. Parampottupadam, K. Parekh. Challenge Participants: K. Kushibar, K. Lekadir, M. Jiang, Y. Yin, Hongzheng Yang, Q. Liu, C. Chen, Q. Dou, P. Heng, X. Zhang, S. Zhang, M. Khan, M. Azeem, M. Jafaritadi, E. Alhoniemi, E. Kontio, S. Khan, L. Mächler, I. Ezhov, F. Kofler, S. Shit, J. Paetzold, T. Loehr, B. Wiestler, H. Peiris, K. Pawar, S. Zhong, Z. Chen, M. Hayat, G. Egan, M. Harandi, E. Polat, G. Polat, A. Kocyigit, A. Temizel, A. Tuladhar, L. Tyagi, R. Souza, N. Forkert, P. Mouches, M. Wilms, V. Shambhat, A. Maurya, S. Danannavar, R. Kalla, V. Anand, G. Krishnamurthi, S. Nalawade, C. Ganesh, B. Wagner, D. Reddy, Y. Das, F. Yu, B. Fei, A. Madhuranthakam, J. Maldjian, G. Singh, J. Ren, W. Zhang, N. An, Q. Hu, Y. Zhang, Y. Zhou, V. Siomos, G. Tarroni, J. Passerrat-Palmbach, A. Rawat, G. Zizzo, S. Kadhe, J. Epperlein, S. Braghin, Y. Wang, R. Kanagavelu, Q. Wei, Y. Yang, Y. Liu, K. Kotowski, S. Adamski, B. Machura, W. Malara, L. Zarudzki, J. Nalepa, Y. Shi, H. Gao, S. Avestimehr, Y. Yan, A. Akbar, E. Kondrateva, Hua Yang, Z. Li, H. Wu, J. Roth, C. Saueressig, A. Milesi, Q. Nguyen, N. Gruenhagen, T. Huang, J. Ma, H. Singh, N. Pan, D. Zhang, R. Zeineldin, M. Futrega, Y. Yuan, G. Conte, X. Feng, Q. Pham, Y. Xia, Z. Jiang, H. Luu, M. Dobko, A. Carré, B. Tuchinov, H. Mohy-ud-Din, S. Alam, A. Singh, N. Shah, W. Wang. Data Contributors: C. Sako, M. Bilello, S. Ghodasara, S. Mohan, C. Davatzikos, E. Calabrese, J. Rudie, J. Villanueva-Meyer, S. Cha, C. Hess, J. Mongan, M.

Ingalhalikar, M. Jadhav, U. Pandey, J. Saini, R. Huang, K. Chang, M. To, S. Bhardwaj, C. Chong, M. Agzarian, M. Kozubek, F. Lux, J. Michálek, P. Matula, M. Ker^kovský, T. Kopr^ivová, M. Dostál, V. Vybíhal, M. Pinho, J. Holcomb, M. Metz, R. Jain, M. Lee, Y. Lui, P. Tiwari, R. Verma, R. Bareja, I. Yadav, J. Chen, N. Kumar, Y. Gusev, K. Bhuvaneshwar, A. Sayah, C. Bencheqroun, A. Belouali, S. Madhavan, R. Colen, A. Kotrotsou, P. Voll-muth, G. Brugnara, C. Preetha, F. Sahm, M. Bendszus, W. Wick, A. Mahajan, C. Balaña, J. Capellades, J. Puig, Y. Choi, S. Lee, J. Chang, S. Ahn, H. Shaykh, A. Herrera-Trujillo, M. Trujillo, W. Escobar, A. Abello, J. Bernal, J. Gómez, P. LaMontagne, D. Marcus, M. Milchenko, A. Nazeri, B. Landman, K. Ramadass, K. Xu, S. Chotai, L. Chambless, A. Mistry, R. Thompson, A. Srinivasan, J. Bapuraj, A. Rao, N. Wang, O. Yoshiaki, T. Moritani, S. Turk, J. Lee, S. Prabhudesai, J. Garrett, M. Larson, R. Jeraj, H. Li, T. Weiss, M. Weller, A. Bink, B. Pouymayou, S. Sharma, T. Tseng, S. Adabi, A. Falcão, S. Martins, B. Teixeira, F. Sprenger, D. Menotti, D. Lucio, S. Niclou, O. Keunen, A. Hau, E. Pelaez, H. Franco-Maldonado, F. Loayza, S. Quevedo, R. McKinley, J. Slotboom, P. Radojewski, R. Meier, R. Wiest, J. Trenkler, J. Pichler, G. Necker. Challenge Organizing Team: M. Zenk, U. Baid, S. Pati, A. Linardos, B. Edwards, M. Sheller, P. Foley, A. Aristizabal, D. Zimmerer, A. Gruzdev, J. Martin, R. Shinohara, A. Reinke, F. Isensee, S. Parampottupadam, K. Parekh, R. Floca, H. Kassem, B. Baheti, S. Thakur, V. Chung, L. Maier-Hein, J. Albrecht, P. Mattson, A. Karargyris, P. Shah, B. Menze, K. Maier-Hein, S. Bakas, Writing the original manuscript: M. Zenk, U. Baid, S. Pati, A. Linardos, K. Maier-Hein, S. Bakas Review, edit, & approval of the final manuscript: All authors.

## Competing interests

The Intel-affiliated authors (B. Edwards, M. Sheller, P. Foley, A. Gruzdev, J. Martin, P. Shah) would like to disclose the following (potential) competing interests as Intel employees. Intel may develop proprietary software that is related in reputation to the OpenFL open source project highlighted in this work. In addition, the work demonstrates feasibility of federated learning for brain tumor boundary detection models. Intel may benefit by selling products to support an increase in demand for this use-case. The remaining authors declare no competing interests.

## Additional information

Maximilian Zenk [1,2,219], Ujjwal Baid [3,4,219], Sarthak Pati [3,4,5], Akis Linardos [3,4], Brandon Edwards [6], Micah Sheller [5,6], Patrick Foley [6], Alejandro Aristizabal [5,7], David Zimmerer [1], Alexey Gruzdev [6], Jason Martin [6], Russell T. Shinohara [8,9,10], Annika Reinke [11,12], Fabian Isensee [1,12], Santhosh Parampottupadam [1], Kaushal Parekh [1], Ralf Floca [1], Hasan Kassem [5], Bhakti Baheti [4], Siddhesh Thakur [4], Verena Chung [13], Kaisar Kushibar [14], Karim Lekadir [15,16], Meirui Jiang [17], Youtan Yin [18], Hongzheng Yang [19], Quande Liu [17], Cheng Chen [17], Qi Dou [17], Pheng-Ann Heng [17], Xiaofan Zhang [20], Shaoting Zhang [21], Muhammad Irfan Khan [22], Mohammad Ayyaz Azeem [23], Mojtaba Jafaritadi [22,24], Esa Alhoniemi [22], Elina Kontio [22], Suleiman A. Khan [25], Leon Mächler [26], Ivan Ezhov [27,28], Florian Kofler [27,28,29,30], Suprosanna Shit [27,28,30], Johannes C. Paetzold [31,32], Timo Loehr [27,28], Benedikt Wiestler [29], Himashi Peiris [33,34,35], Kamlesh Pawar [33,36], Shenjun Zhong [33,37], Zhaolin Chen [33,35], Munawar Hayat [35], Gary Egan [33,36], Mehrtash Harandi [34], Ece Isik Polat [38], Gorkem Polat [38], Altan Kocyigit [38], Alptekin Temizel [38], Anup Tuladhar [39,40], Lakshay Tyagi [41], Raissa Souza [39,40,42], Nils D. Forkert [39,40,43,44], Pauline Mouches [39,40,42], Matthias Wilms [39,40], Vishruth Shambhat [45], Akansh Maurya [46], Shubham Subhas Danannavar [45], Rohit Kalla [45], Vikas Kumar Anand [45], Ganapathy Krishnamurthi [45], Sahil Nalawade [47], Chandan Ganesh [47], Ben Wagner [47], Divya Reddy [47], Yudhajit Das [47], Fang F. Yu [47], Baowei Fei [48], Ananth J. Madhuranthakam [47], Joseph Maldjian [47], Gaurav Singh [49], Jianxun Ren [50], Wei Zhang [50], Ning An [50], Qingyu Hu [51], Youjia Zhang [50], Ying Zhou [50], Vasilis Siomos [52], Giacomo Tarroni [52,53], Jonathan Passerrat-Palmbach [52,53], Ambrish Rawat [54], Giulio Zizzo [54], Swanand Ravindra Kadhe [54], Jonathan P. Epperlein [54], Stefano Braghin [54], Yuan Wang [55], Renuga Kanagavelu [55], Qingsong Wei [55], Yechao Yang [55], Yong Liu [55], Krzysztof Kotowski [56], Szymon Adamski [56], Bartosz Machura [56], Wojciech Malara [56], Lukasz Zarudzki [57], Jakub Nalepa [56,58], Yaying Shi [59,60], Hongjian Gao [61], Salman Avestimehr [61], Yonghong Yan [59], Agus S. Akbar [62], Ekaterina Kondrateva [63], Hua Yang [64], Zhaopei Li [65], Hung-Yu Wu [66], Johannes Roth [67], Camillo Saueressig [68], Alexandre Milesi [69], Quoc D. Nguyen [70], Nathan J. Gruenhagen [71], Tsung-Ming Huang [72], Jun Ma [73], Har Shwinder H. Singh [74], Nai-Yu Pan [75], Dingwen Zhang [76], Ramy A. Zeineldin [77], Michal Futrega [69], Yading Yuan [78,79], Gian Marco Conte [80], Xue Feng [81], Quan D. Pham [82], Yong Xia [83], Zhifan Jiang [84],

Huan Minh Luu[85], Mariia Dobko[86], Alexandre Carré[87], Bair Tuchinov[88], Hassan Mohy-ud-Din[89], Saruar Alam[90], Anup Singh[91], Nameeta Shah [92], Weichung Wang[93], Chiharu Sako [8,94], Michel Bilello[8,94], Satyam Ghodasara [94], Suyash Mohan [8,94], Christos Davatzikos [8,94], Evan Calabrese [95], Jeffrey Rudie[95], Javier Villanueva-Meyer[95], Soonmee Cha[95], Christopher Hess [95], John Mongan[95], Madhura Ingalhalikar[96], Manali Jadhav [96], Umang Pandey [96], Jitender Saini[97], Raymond Y. Huang[98], Ken Chang[99], Minh-Son To[100,101], Sargam Bhardwaj[100], Chee Chong[101], Marc Agzarian [100,101], Michal Kozubek [102], Filip Lux [102], Jan Michálek [102], Petr Matula [102], Miloš Ker^kovský [103], Tereza Kopr^ivová [103], Marek Dostál [103,104], Václav Vybíhal [105], Marco C. Pinho[47], James Holcomb[47], Marie Metz [106], Rajan Jain [107,108], Matthew D. Lee [107], Yvonne W. Lui [107], Pallavi Tiwari[109,110], Ruchika Verma[111,112,113], Rohan Bareja[111], Ipsa Yadav[111], Jonathan Chen [111], Neeraj Kumar[114,115,116], Yuriy Gusev[117], Krithika Bhuvaneshwar [117], Anousheh Sayah [118], Camelia Benchequroun[117], Anas Belouali[117], Subha Madhavan[117], Rivka R. Colen[119,120], Aikaterini Kotrotsou[120], Philipp Vollmuth[1,121,122], Gianluca Brugnara [123], Chandrakanth J. Preetha [123], Felix Sahm [124,125], Martin Bendszus[123], Wolfgang Wick [2,126], Abhishek Mahajan [127,128], Carmen Balaña [129], Jaume Capellades[130], Josep Puig[131], Yoon Seong Choi [132], Seung-Koo Lee [133], Jong Hee Chang [133], Sung Soo Ahn [133], Hassan F. Shaykh [134], Alejandro Herrera-Trujillo [135,136], Maria Trujillo[136], William Escobar[135], Ana Abello[136], Jose Bernal [137,138,139], Jhon Gómez[136], Pamela LaMontagne [140], Daniel S. Marcus[140], Mikhail Milchenko [140,141], Arash Nazeri [140], Bennett Landman [142], Karthik Ramadass [142], Kaiwen Xu [143], Silky Chotai [144], Lola B. Chambless[144], Akshitkumar Mistry[144], Reid C. Thompson[144], Ashok Srinivasan[145], J. Rajiv Bapuraj [145], Arvind Rao[146], Nicholas Wang [146], Ota Yoshiaki[145], Toshio Moritani[145], Sevcan Turk[145], Joonsang Lee[146], Snehal Prabhudesai[146], John Garrett [147,148], Matthew Larson[147], Robert Jeraj[148], Hongwei Li[30], Tobias Weiss [149], Michael Weller [149], Andrea Bink [150], Bertrand Pouymayou [150], Sonam Sharma[151], Tzu-Chi Tseng[151], Saba Adabi[151], Alexandre Xavier Falcão [152], Samuel B. Martins [153], Bernardo C. A. Teixeira [154,155], Flávia Sprenger [155], David Menotti [156], Diego R. Lucio[156], Simone P. Niclou [157,158], Olivier Keunen [159], Ann-Christin Hau [157,160], Enrique Pelaez [161], Heydy Franco-Maldonado [162], Francis Loayza [161], Sebastian Quevedo [163], Richard McKinley [164], Johannes Slotboom [164], Piotr Radojewski[164], Raphael Meier[164], Roland Wiest [164,165], Johannes Trenkler [166], Josef Pichler[167], Georg Necker[166], Andreas Haunschmidt [166], Stephan Meckel [166,168], Pamela Guevara [169], Esteban Torche[169], Cristobal Mendoza [169], Franco Vera [169], Elvis Ríos [169], Eduardo López [169], Sergio A. Velastin[170,171], Joseph Choi[172], Stephen Baek[173], Yusung Kim[174], Heba Ismael[174], Bryan Allen [174], John M. Buatti [174], Peter Zampakis [175], Vasileios Panagiotopoulos [176], Panagiotis Tsiganos [177], Sotiris Alexiou [178], Ilias Haliassos [179], Evangelia I. Zacharaki [178], Konstantinos Moustakas [178], Christina Kalogeropoulou [175], Dimitrios M. Kardamakis[179], Bing Luo[180], Laila M. Poisson [181], Ning Wen[180], Martin Vallières [182,183], Mahdi Ait Lhaj Loutfi [182], David Fortin[184], Martin Lepage [185], Fanny Morón [186], Jacob Mandel [187], Gaurav Shukla[8,188,189], Spencer Liem[190], Gregory S. Alexandre[190,191], Joseph Lombardo[189,190], Joshua D. Palmer [192], Adam E. Flanders[193], Adam P. Dicker[189], Godwin Ogbole[194], Dotun Oyekunle [194], Olubunmi Odafe-Oyibotha[195], Babatunde Osobu[194], Mustapha Shu'aibu Hikima[196], Mayowa Soneye[194], Farouk Dako[94], Adeleye Dorcas[197], Derrick Murcia[198], Eric Fu[198], Rourke Haas[198], John A. Thompson [199], David Ryan Ormond [198], Stuart Currie [200], Kavi Fatania [200], Russell Frood [200], Amber L. Simpson [201,202], Jacob J. Peoples [201], Ricky Hu[201,202], Danielle Cutler [201,202,203,204], Fabio Y. Moraes[205], Anh Tran [201,202], Mohammad Hamghalam[201,206], Michael A. Boss [207], James Gimpel [207], Deepak Kattil Veettil [208], Kendall Schmidt[208], Lisa Cimino[208], Cynthia Price[208], Brian Bialecki [208], Sailaja Marella[208], Charles Apgar[207], Andras Jakab [209], Marc-André Weber [210], Errol Colak[211], Jens Kleesiek [212], John B. Freymann[213], Justin S. Kirby[213], Lena Maier-Hein[11], Jake Albrecht[13], Peter Mattson [5], Alexandros Karargyris [5], Prashant Shah[6], Bjoern Menze [27,28,30], Klaus Maier-Hein[1,2,12,214,220] & Spyridon Bakas [3,4,5,215,216,217,218,220] ✉

[1]German Cancer Research Center (DKFZ) Heidelberg, Division of Medical Image Computing, Heidelberg, Germany. [2]Medical Faculty Heidelberg, Heidelberg University, Heidelberg, Germany. [3]Center for Federated Learning in Medicine, Indiana University, Indianapolis, IN, USA. [4]Division of Computational Pathology, Department of Pathology and Laboratory Medicine, Indiana University School of Medicine, Indianapolis, IN, USA. [5]Medical Research Group, MLCommons, San Francisco, CA, USA. [6]Intel Corporation, Santa Clara, CA, USA. [7]Factored, Palo Alto, CA, USA. [8]Center for AI and Data Science for Integrated Diagnostics (AI2D), University of Pennsylvania, Philadelphia, PA, USA. [9]Department of Biostatistics, Epidemiology and Informatics, Perelman School of Medicine, University of Pennsylvania, Philadelphia, PA, USA. [10]Penn Statistics in Imaging and Visualization Center, Perelman School of Medicine, University of Pennsylvania, Philadelphia, PA, USA. [11]German Cancer Research Center (DKFZ) Heidelberg, Division of Intelligent Medical Systems, Heidelberg, Germany. [12]Helmholtz Imaging, German Cancer Research Center (DKFZ), Heidelberg, Germany. [13]Sage Bionetworks, Seattle, WA, USA. [14]Department of Mathematics and Computer Science, Universitat de Barcelona, Barcelona, Spain. [15]Department of Mathematics and Computer Science, Universitat de Barcelona, Artificial Intelligence in Medicine Lab (BCN-AIM), Barcelona, Spain. [16]Institució Catalana de Recerca i Estudis Avançats (ICREA), Barcelona, Spain. [17]Department of Computer Science and Engineering, The Chinese University of Hong Kong, Hong Kong SAR, China. [18]Department of Computer Science and Technology, Zhejiang University, Hangzhou, China. [19]Department of Computer Science and Engineering, Beihang University, Beijing, China. [20]Shanghai Jiao Tong University, Shanghai, China. [21]Shanghai Artificial Intelligence Laboratory, Shanghai, China. [22]School of Data Engineering and AI Technologies, Turku University of Applied Sciences,

Turku, Finland. [23]Riphah International University, Islamabad, Pakistan. [24]Department of Radiology, Stanford University, Stanford, CA, USA. [25]University of Helsinki, Helsinki, Finland. [26]École Normale Supérieure, Paris, France. [27]Department of Informatics, Technical University of Munich, Munich, Germany. [28]TranslaTUM - Central Institute for Translational Cancer Research, Technical University of Munich, Munich, Germany. [29]Department of Diagnostic and Interventional Neuroradiology, School of Medicine and Health, Technical University of Munich, Munich, Germany. [30]Department of Quantitative Biomedicine, University of Zurich, Zurich, Switzerland. [31]ITERM Institute Helmholtz Zentrum Muenchen, Neuherberg, Germany. [32]Department of Radiology, Weill Cornell Medicine, Cornell University, New York, NY, USA. [33]Monash Biomedical Imaging, Monash University, Melbourne, VIC, Australia. [34]Department of Electrical and Computer Systems Engineering, Faculty of Engineering, Monash University, Melbourne, VIC, Australia. [35]Department of Data Science and AI, Faculty of Information Technology, Monash University, Melbourne, VIC, Australia. [36]School of Psychological Sciences, Monash University, Melbourne, VIC, Australia. [37]National Imaging Facility, St Lucia, QLD, Australia. [38]Graduate School of Informatics, Middle East Technical University, Ankara, Turkey. [39]Department of Radiology, Cumming School of Medicine, University of Calgary, Calgary, AB, Canada. [40]Hotchkiss Brain Institute, University of Calgary, Calgary, AB, Canada. [41]Department of Chemical Engineering, Indian Institute of Technology Kanpur, Kanpur, Uttar Pradesh, India. [42]Biomedical Engineering Program, University of Calgary, Calgary, AB, Canada. [43]Department of Clinical Neurosciences, Cumming School of Medicine, University of Calgary, Calgary, AB, Canada. [44]Alberta Children's Hospital Research Institute, University of Calgary, Calgary, AB, Canada. [45]Department of Engineering Design, IIT Madras, Chennai, India. [46]Robert Bosch Center of Data Science and AI, IIT Madras, Chennai, India. [47]University of Texas Southwestern Medical Center, Dallas, TX, USA. [48]Department of Bioengineering, University of Texas at Dallas, Dallas, TX, USA. [49]Indian Institute of Information Technology Vadodara, Gandhinagar, India. [50]Changping Laboratory, Beijing, China. [51]School of Electrical and Computer Engineering, Cornell University, Ithaca, NY, USA. [52]City St George's, University of London, London, UK. [53]Imperial College London, London, UK. [54]IBM Research, Dublin, Ireland. [55]Institute of High Performance Computing (IHPC), Agency for Science, Technology and Research (A*STAR), Singapore, Singapore. [56]Graylight Imaging, Gliwice, Poland. [57]Maria Sklodowska-Curie Memorial Cancer Center and Institute of Oncology, Gliwice, Poland. [58]Silesian University of Technology, Gliwice, Poland. [59]University of North Carolina at Charlotte, Charlotte, NC, USA. [60]Department of Radiation Physics, The University of Texas MD Anderson Cancer Center, Houston, TX, USA. [61]University of Southern California, Los Angeles, CA, USA. [62]Universitas Islam Nahdlatul Ulama Jepara, Jepara, Indonesia. [63]Skolkovo Institute of Science and Technology, Moscow, Russia. [64]Fujian Normal University, Fuzhou, China. [65]Fuzhou University, Fuzhou, China. [66]National Tsing Hua University, Hsinchu, Taiwan. [67]ScaDS.AI, Dresden, Germany. [68]Brown University, Providence, RI, USA. [69]NVIDIA, Santa Clara, CA, USA. [70]EPITA, Le Kremlin-Bicêtre, France. [71]Medical College of Wiconsin, Milwaukee, WI, USA. [72]Department of Mathematics, National Taiwan Normal University, Taipei, Taiwan. [73]Nanjing University of Science and Technology, Nanjing, China. [74]Hong Kong University of Science and Technology, Hong Kong, Hong Kong Special Administrative Region of China, China. [75]National Taiwan University of Science and Technology, Taipei, Taiwan. [76]School of Automation, Northwestern Polytechnical University, Xi'an, China. [77]Institute for Anthropomatics and Robotics, Karlsruhe Institute of Technology, Karlsruhe, Germany. [78]Department of Radiation Oncology, Columbia University Irving Medical Center, New York, NY, USA. [79]Columbia University Data Science Institute, New York, NY, USA. [80]Department of Radiology, Mayo Clinic, Rochester, MN, USA. [81]University of Virginia, Charlottesville, VA, USA. [82]VinBrain, Hanoi, Vietnam. [83]School of Computer Science and Engineering, Northwestern Polytechnical University, Xi'an, China. [84]Children's National Hospital, Washington, DC, USA. [85]MRI Lab, KAIST, Daejeon, Korea. [86]Ukrainian Catholic University, Lviv, Ukraine. [87]Gustave Roussy Cancer Campus, Villejuif, France. [88]Novosibirsk State University, Novosibirsk, Russia. [89]Department of Electrical Engineering, Syed Babar Ali School of Science and Engineering, LUMS, Lahore, Pakistan. [90]University of Bergen, Bergen, Norway. [91]Indian Institute of Technology, Delhi, India. [92]Mazumdar Shaw Medical Foundation, Bengaluru, India. [93]Institute of Applied Mathematical Sciences, National Taiwan University, Taipei, Taiwan. [94]Department of Radiology, Perelman School of Medicine at the University of Pennsylvania, Philadelphia, PA, USA. [95]Department of Radiology and Biomedical Imaging, University of California San Francisco, San Francisco, CA, USA. [96]Symbiosis Center for Medical Image Analysis, Symbiosis International University, Pune, India. [97]Department of Neuroimaging and interventional Radiology, National Institute of Mental Health and Neurosciences, Bangalore, India. [98]Department of Radiology, Brigham and Women's Hospital, Harvard Medical School, Boston, MA, USA. [99]Athinoula A. Martinos Center for Biomedical Imaging, Massachusetts General Hospital, Charlestown, MA, USA. [100]College of Medicine and Public Health, Flinders University, Bedford Park, SA, Australia. [101]South Australia Medical Imaging, Flinders Medical Centre, Bedford Park, SA, Australia. [102]Centre for Biomedical Image Analysis, Faculty of Informatics, Masaryk University, Brno, Czech Republic. [103]Department of Radiology and Nuclear Medicine, University Hospital Brno and Faculty of Medicine, Masaryk University, Brno, Czech Republic. [104]Department of Biophysics, Faculty of Medicine, Masaryk University, Brno, Czech Republic. [105]Department of Neurosurgery, University Hospital Brno and Faculty of Medicine, Masaryk University, Brno, Czech Republic. [106]Department of Neuroradiology, Klinikum rechts der Isar, Munich, Germany. [107]Department of Radiology, NYU Grossman School of Medicine, New York, NY, USA. [108]Department of Neurosurgery, NYU Grossman School of Medicine, New York, NY, USA. [109]Department of Radiology, Biomedical Engineering, Medical Physics, University of Wisconsin School of Medicine & Public Health, Madison, WI, USA. [110]William S. Middleton Memorial Veterans Affairs (VA), Madison, WI, USA. [111]Case Western Reserve University, Cleveland, OH, USA. [112]Windreich Department of Artificial Intelligence and Human Health, Icahn School of Medicine at Mount Sinai, New York, NY, USA. [113]Hasso Plattner Institute for Digital Health at Mount Sinai, Icahn School of Medicine at Mount Sinai, New York, NY, USA. [114]University of Alberta, Edmonton, AB, Canada. [115]Alberta Machine Intelligence Institute, Edmonton, AB, Canada. [116]Department of Pathology, Memorial Sloan Kettering Cancer Center, New York, NY, USA. [117]Innovation Center for Biomedical Informatics (ICBI), Georgetown University, Washington, DC, USA. [118]Division of Neuroradiology & Neurointerventional Radiology, MedStar Georgetown University Hospital, Department of Radiology, Washington, DC, USA. [119]Department of Radiology, Neuroradiology Division, University of Pittsburgh, Pittsburgh, PA, USA. [120]Department of Diagnostic Radiology, University of Texas MD Anderson Cancer Center, Houston, TX, USA. [121]Division for Computational Radiology & Clinical AI, University Hospital Bonn, Bonn, Germany. [122]Faculty of Medicine, University of Bonn, Bonn, Germany. [123]Department of Neuroradiology, Heidelberg University Hospital, Heidelberg, Germany. [124]Department of Neuropathology, Heidelberg University Hospital, Heidelberg, Germany. [125]Clinical Cooperation Unit Neuropathology, German Cancer Consortium (DKTK) within the German Cancer Research Center (DKFZ), Heidelberg, Germany. [126]Neurology Clinic, Heidelberg University Hospital, Heidelberg, Germany. [127]Department of Radiodiagnosis and Imaging, Tata Memorial Centre, Tata Memorial Hospital, HBNI, Mumbai, India. [128]Department of Imaging, The Clatterbridge Cancer Centre NHS Foundation Trust, Liverpool, UK. [129]Catalan Institute of Oncology, Badalona, Spain. [130]Consorci MAR Parc de Salut de Barcelona, Catalonia, Spain. [131]Radiology Department CDI and IDIBAPS, Hospital Clinic of Barcelona, Barcelona, Spain. [132]National University of Singapore, Yong Loo Lin School of Medicine, Singapore, Singapore. [133]Yonsei University College of Medicine, Seoul, Korea. [134]University of Alabama in Birmingham, Birmingham, AL, USA. [135]Clínica Imbanaco QuirónSalud, Cali, Colombia. [136]Universidad del Valle, Cali, Colombia. [137]The University of Edinburgh, Edinburgh, UK. [138]German Centre for Neurodegenerative Diseases (DZNE), Magdeburg, Germany. [139]Institute for cognitive neurology and dementia research (IKND), Magdeburg, Germany. [140]Department of Radiology, Washington University in St. Louis, St. Louis, MO, USA. [141]Neuroimaging Informatics and Analysis Center, Washington University in St. Louis, St. Louis, MO, USA. [142]Department of Electrical and Computer Engineering, Vanderbilt University, Nashville, TN, USA. [143]Department of Computer Science, Vanderbilt University, Nashville, TN, USA. [144]Department of Neurosurgery, Vanderbilt University Medical Center, Nashville, TN, USA. [145]Department of Neuroradiology, University of Michigan, Ann Arbor, MI, USA. [146]Department of Computational Medicine and Bioinformatics, University of Michigan, Ann Arbor, MI, USA.

[147]Department of Radiology, UW Madison School of Medicine and Public Health, University of Wisconsin, Madison, WI, USA. [148]Department of Medical Physics, UW Madison School of Medicine and Public Health, University of Wisconsin, Madison, WI, USA. [149]Department of Neurology and Clinical Neuroscience Center, University Hospital Zurich and University of Zurich, Zurich, Switzerland. [150]Department of Neuroradiology and Clinical Neuroscience Center, University Hospital Zurich and University of Zurich, Zurich, Switzerland. [151]Department of Radiation Oncology, Icahn School of Medicine at Mount Sinai, New York, NY, USA. [152]Institute of Computing, University of Campinas, Campinas, São Paulo, Brazil. [153]Federal Institute of Education, Science, and Technology of São Paulo, Araraquara, São Paulo, Brazil. [154]Instituto de Neurologia de Curitiba, Curitiba, Paraná, Brazil. [155]Federal University of Parana, Curitiba, Paraná, Brazil. [156]Department of Informatics, Federal University of Parana, Curitiba, Paraná, Brazil. [157]NORLUX Neuro-Oncology Laboratory, Luxembourg Institute of Health, Luxembourg, Luxembourg. [158]Faculty of Science, Technology and Medicine, University of Luxembourg, Esch-sur-Alzette, Luxembourg. [159]Brain Imaging and Neuro Epidemiology Group, Luxembourg Institute of Health, Luxembourg, Luxembourg. [160]Goethe University, University Hospital, Dr. Senckenberg Institute of Neurooncology, Frankfurt am Main, Germany. [161]Escuela Superior Politecnica del Litoral, Guayaquil, Guayas, Ecuador. [162]Sociedad de Lucha Contral el Cancer - SOLCA, Guayaquil, Ecuador. [163]Universidad Católica de Cuenca, Cuenca, Ecuador. [164]Support Center for Advanced Neuroimaging, University Institute of Diagnostic and Interventional Neuroradiology, University Hospital Bern, Inselspital, University of Bern, Bern, Switzerland. [165]Institute for Surgical Technology and Biomechanics, University of Bern, Bern, Switzerland. [166]Institute of Neuroradiology, Neuromed Campus (NMC), Kepler University Hospital Linz, Linz, Austria. [167]Department of Neurooncology, Neuromed Campus (NMC), Kepler University Hospital Linz, Linz, Austria. [168]Institute of Diagnostic and Interventional Neuroradiology, RKH Klinikum Ludwigsburg, Ludwigsburg, Germany. [169]Universidad de Concepción, Concepción, Chile. [170]School of Electronic Engineering and Computer Science, Queen Mary University of London, London, UK. [171]Department of Computer Engineering, Universidad Carlos III de Madrid, Madrid, Spain. [172]Department of Industrial and Systems Engineering, University of Iowa, Iowa City, IA, USA. [173]Department of Industrial and Systems Engineering, Department of Radiation Oncology, University of Iowa, Iowa City, IA, USA. [174]Department of Radiation Oncology, University of Iowa, Iowa City, IA, USA. [175]Department of NeuroRadiology, University of Patras, Patras, Greece. [176]Department of Neurosurgery, University of Patras, Patras, Greece. [177]Clinical Radiology Laboratory, Department of Medicine, University of Patras, Patras, Greece. [178]Department of Electrical and Computer Engineering, University of Patras, Patras, Greece. [179]Department of Radiation Oncology, University of Patras, Patras, Greece. [180]Department of Radiation Oncology, Henry Ford Health, Detroit, MI, USA. [181]Department of Public Health Sciences, Henry Ford Health, Detroit, MI, USA. [182]Department of Computer Science, Université de Sherbrooke, Sherbrooke, QC, Canada. [183]Centre de recherche du Centre hospitalier universitaire de Sherbrooke, Sherbrooke, QC, Canada. [184]Division of Neurosurgery and Neuro-Oncology, Faculty of Medicine and Health Science, Université de Sherbrooke, Sherbrooke, QC, Canada. [185]Department of Nuclear Medicine and Radiobiology, Sherbrooke Molecular Imaging Centre, Université de Sherbrooke, Sherbrooke, QC, Canada. [186]Department of Radiology, Baylor College of Medicine, Houston, TX, USA. [187]Department of Neurology, Baylor College of Medicine, Houston, TX, USA. [188]Department of Radiation Oncology, Christiana Care Health System, Philadelphia, PA, USA. [189]Department of Radiation Oncology, Sidney Kimmel Comprehensive Cancer Center, Thomas Jefferson University, Philadelphia, PA, USA. [190]Sidney Kimmel Medical College, Thomas Jefferson University, Philadelphia, PA, USA. [191]Department of Radiation Oncology, University of Maryland, Baltimore, MD, USA. [192]Department of Radiation Oncology, James Cancer Center, The Ohio State University, Columbus, OH, USA. [193]Department of Radiology, Sidney Kimmel Cancer Center, Thomas Jefferson University, Philadelphia, PA, USA. [194]Department of Radiology, University College Hospital Ibadan, Oyo, Nigeria. [195]Clinix Healthcare, Lagos, Lagos, Nigeria. [196]Department of Radiology, Muhammad Abdullahi Wase Teaching Hospital, Kano, Nigeria. [197]Department of Radiology, Obafemi Awolowo University Ile-Ife, Ile-Ife, Osun, Nigeria. [198]Department of Neurosurgery, University of Colorado Anschutz Medical Campus, Aurora, CO, USA. [199]Departments of Neurosurgery and Neurology, University of Colorado Anschutz Medical Campus, Aurora, CO, USA. [200]Department of Radiology, Leeds Teaching Hospitals Trust, Leeds, UK. [201]School of Computing, Queen's University, Kingston, ON, Canada. [202]Department of Biomedical and Molecular Sciences, Queen's University, Kingston, ON, Canada. [203]Faculty of Medicine and Health Sciences, McGill University, Montreal, QC, Canada. [204]Faculty of Arts and Sciences, Queen's University, Kingston, ON, Canada. [205]Department of Oncology, Queen's University, Kingston, ON, Canada. [206]Department of Electrical Engineering, Qazvin Branch, Islamic Azad University, Qazvin, Iran. [207]Center for Research and Innovation, American College of Radiology, Philadelphia, PA, USA. [208]American College of Radiology (ACR), Reston, VA, USA. [209]Center for MR-Research, University Children's Hospital Zurich, Zurich, Switzerland. [210]Institute of Diagnostic and Interventional Radiology, Pediatric Radiology and Neuroradiology, University Medical Center Rostock, Rostock, Germany. [211]Department of Medical Imaging, Unity Health Toronto, University of Toronto, Toronto, ON, Canada. [212]Institute for AI in Medicine (IKIM), University Hospital Essen, Essen, Germany. [213]Leidos Biomedical Research, Inc., Frederick National Laboratory for Cancer Research, Frederick, MD, USA. [214]Pattern Analysis and Learning Group, Department of Radiation Oncology, Heidelberg University Hospital, Heidelberg, Germany. [215]Indiana University Melvin and Bren Simon Comprehensive Cancer Center, Indianapolis, IN, USA. [216]Department of Radiology and Imaging Sciences, Indiana University School of Medicine, Indianapolis, IN, USA. [217]Department of Neurological Surgery, Indiana University School of Medicine, Indianapolis, IN, USA. [218]Department of Computer Science, Luddy School of Informatics, Computing, and Engineering, Indiana University, Indianapolis, IN, USA. [219]These authors contributed equally: Maximilian Zenk, Ujjwal Baid. [220]These authors jointly supervised this work: Klaus Maier-Hein, Spyridon Bakas. ✉e-mail: spbakas@iu.edu

