## [Peer Review File · Nature Communications]

Towards Fair Decentralized Benchmarking of Healthcare AI Algorithms with the Federated Tumor Segmentation (FeTS) Challenge

Corresponding Author: Dr Spyridon Bakas

Figure on page 23 in this Peer Review File has been amended to remove third-party material where no permission to publish could be obtained.

Version 0:

Reviewer comments:

Reviewer #1

(Remarks to the Author)

This work is of significant merit as it is “the very first to establish a decentralized algorithmic benchmarking paradigm for both federated learning (FL) and federated evaluation (FE)”. It is particularly noteworthy for constructing a large-scale FE benchmark incorporating data from 32 globally distributed sites and benchmarking 41 models. Considerable effort has been dedicated to developing this innovative FE benchmark, which could serve as a foundational step towards creating generalizable algorithms for cross-center applications. Therefore, the significance of this work certainly merits recognition.

I have the following concerns:

1. In the FE task, only 5 out of 35 registered teams submit results for this task. It is regrettable that the few submissions (only 5) did not yield many intriguing conclusions, and models from BraTS that were not specifically optimized for FE testing still ranked at the top. What's the main reason for the little engagement? What can be done to improve the submission rate?
2. In the FE task, the evaluation currently requires accessing data stored on each site's server. How can this benchmark be maintained in the long term after the challenge ends, so that the following research can continue and benefit from it? Alternatively, how much test data can be accessible to the community after overcoming privacy concerns?
3. In the FL task, introducing communication efficiency in evaluation is reasonable. However, the details of normalization on hardware systems, as mentioned in line 921 “To calculate this metric, we simulate the time taken to run each round, and it is normalized across a single hardware system, so that final scoring will not depend on the hardware used.”, were not clarified.
4. In the FL task, can you provide more details on how the organizers simulated federated data based on the training data of BraTS21?
5. In the FE task, information on data distribution of different sites (e.g., age, ethnicity, vendor/manufacture) is lacking, which would facilitate understanding the challenges of generalizability in this benchmark, and potentially help the development of algorithms with better generalizability.
6. Considering the federated data is simulated from the BraTS21 dataset, it would be interesting to assess and analyze how the performance of segmentation algorithms developed under FL compares to directly using the BraTS21 dataset. In other words, to what extent can current FL algorithms approach this baseline? This may shed light on the potential and meaning of this FL challenge.
7. The number of image scans across different sites is quite imbalanced (Fig. C8), as well as their graphical regions. Although it shows significant improvement in size and diversity compared to the BraTS dataset, and this imbalance is taken into account in rankings computation, it would be beneficial to alleviate this imbalance through further data collection.
8. A small typo: The reference to section 2.4 in line 1724 of the supplementary materials seems incorrect; perhaps it was intended to refer to section 4.1.3?

(Remarks on code availability)
The code is in good shape.

Reviewer #2

(Remarks to the Author)

Recap:

The reviewer finds this study of great interest as it 1. contains extensive evaluations of models on real-world healthcare data from numerous institutions 2. depicts the practical hurdles associated with doing multi-sites ML with medical institutions, notably very pragmatic considerations such as coordinating partners (e-mails, zoom calls, etc.), IT-setup (reproducibility, hybrid IT servers, etc.), data quality checks wo cheating, which complements nicely findings already reported by [1, 2], 3. inspects closely failure cases of models, which goes further than quantitative metrics such as DICE and are of utmost importance for medical research. It is the belief of the reviewer that ultimately such findings should be published.

However the reviewer finds that there are a few major (and minor) points on which the authors should provide detailed answers before this manuscript can be considered ready for publication.

The reviewer lists those points below.

Note that the reviewer only commented on points related to their domain of expertise.

Major:

1. Although Figure 1 indeed labels federated learning as operating on "simulated federated data", the fact that the federated learning part is only simulated is not put forward enough in the paper. This is a major concern for the reviewer as the reviewer feels some sentences are a bit misleading such as the authors writing in the abstract that their work stand out from related initiatives as "[other related works only] simulat[e] real-world conditions in a controlled environment" l489, which is exactly what the authors themselves are doing for the federated training part. The reviewer emphasize that they do not consider simulating FL as a weakness of the paper as long as it is clear in the paper see i.e. [3], which simulates FL but is interesting nonetheless. Readers even have to wait until Methods to understand "that training on pooled data was allowed in this task" l985 so authors are comparing pooled methods with no restrictions and federated methods with potentially restrictions on the number of rounds or on the "privacy" of communicate updates but even those restrictions are not clear see point 5. The authors' non simulated contribution exists but is on the evaluation part.

2. The claim "the 1st FL challenge conducted" l564 is a bit vague and therefore the reviewer is not sure if it is valid for instance was LEAF in 2018 a challenge [6] ? What is the precise definition of a challenge apart from "Computational competitions" l1 ? Is it to have a private leaderboard ? The reviewer is not sure even with such definition that this work is the first of its kind. The weaker claim that authors do later "[authors are] the first to circulate the solutions provided by challenge participants across multiple collaborating healthcare sites" l595 within a challenge is already strong but truer: it might be true within the context of a challenge but external validation in other medical institutions is standard for ML applications in healthcare and has been widely used by most works published in nature-type journals. However the reviewer agrees that the way the authors do it is novel (providing docker images + QC instructions).

3. Related to point 2. the term "federated evaluation" which the authors use multiple times is not well-defined nor standard in the community (the 2 references the authors provide using the term "federated evaluation" 77 and 115 in the paper are a somehow cherry-picked and have very few citations). The reviewer understands that authors use that term to describe the reproducible docker image that authors send to different medical institutions and the associated testing protocol, which are very interesting by themselves but 1. it should be clearly defined in the text what it means and 2. the use of the "federated" adjective is in the opinion of the reviewer misleading. The reviewer strongly encourages authors to use clearer naming such as "reproducible multi-sites evaluation protocol" or something along those lines as the reviewer understands for instance as non FL methods are tested (see l985) "federated evaluation" is not i.e. the evaluation of federated learning methods.

4. The term "communication costs" is also used frequently by the authors but in a sense that is not clear and potentially different to what is meant in most federated learning literature [4]. In the literature this term is used to describe the total amount of data that is sent by participants to the server or the number of rounds of FL see i.e. [5] which proposes clever sparsification and compression mechanisms on the FL updates. What the authors mean in this work is unclear. Searching for "communication" gives few hits the most relevant one seems line 921 "Communication efficiency is an additional metric used for Task 1. To calculate this metric, we simulate the time taken to run each round, and it is normalized across a single hardware system, so that final scoring will not depend on the hardware used" and l926 "The communication efficiency metric will be computed as the area under the validation learning curve over 1 week of simulated time". While the reviewer agrees there might be a light coupling between the number of parameters sent and this normalized simulated time. How is this computed when methods are pooled ? Is there a penalty being applied ? Can the authors provide a more thorough definition of this metric and how it is computed for FL methods vs pooled methods ? The reviewer understands sampling client plays a role in this ranking but is it not clear to the reviewer why ? Are training runs' times in the different institutions counted sequentially ? Is round time dominated by one participant having lots of training data ? Notably authors seem to also take into account the cost of communication where communication in this case refers to sending e-mails and doing zoom calls "which were fixed with remote support on shared log files, emails and video calls. This resulted in slow feedback loops and revealed communication as a primary bottleneck" l722. Could the authors clarify this point with equations potentially or detailed protocol with an example ?

5. One of the weakest part of this paper is indeed the evaluation setup and the comparison between participating teams. It seems there are very few constraints on what participants can do apart from the ones listed 1961 to 971 and therefore it is difficult for the reviewer to assess if the different approaches all operate on an equal footing. The reviewer thinks some information is missing such as: Is the number of rounds fixed ? In this case how can the pooled method be valid ? How can the test-time batch normalization methods be valid ? While a setting with few constraints might be warranted, submissions should be clustered and compared between roughly similar algorithms and training/testing protocols. i.e. say model ID p to q use true FL, model X to Y use pooled data, model ID X' use cyclic averaging, model ID X'' use test-time batch-normalization adaptation etc. Could the authors provide a results table with such categories ?

6. Related to point 5, it seems that for authors are evaluating mainly aggregation weights strategies leading to variants of federated averaging. While this setup is possible this disregards most of the advances in FL algorithms playing on the vectorized updates themselves using improvements like control variates [6] or penalizing shifts from consensus using elastic losses[7] therefore it should be clearly stated that the evaluation is only comparing the aggregation weighting schemes.

7. In what way does the claim that the challenge "resemble a phase 3 clinical trial" 1594 is justified ? Are the authors performing randomization and causal inference ?

Minor

1. white tiles from Fig 2 indicate missing runs due to "issues related to GPU compatibility" 1735. Why couldn't the python script fall back on CPU if that was the case ? Couldn't the docker image protocol be robustified ?
2. Fig 2. heatmap is not well-chosen as the spectrum between dark colors indicating "bad" performance and lighter colors is difficult to follow the reviewer advises the authors to use other colormaps potentially non continuous if it is simpler to see i.e. from 0.5 to 0.7 DICE: dark red, from 0.7 to 0.8 orange, from 0.8 to 1. green.
3. The reviewer would appreciate seeing in Fig3. how do other models fare ? Is performance ordered similarly across institutions for different models ?
4. Aggregation schemes used by participants should be described in details in Methods. Descriptions like "together with a sophisticated method for selective sampling that models centers using a Poisson distribution" 1664 are insufficient to assess the methods even if results can be found in referenced technical reports.

[1] Pati, S., Baid, U., Edwards, B. et al. Federated learning enables big data for rare cancer boundary detection. *Nat Commun* 13, 7346 (2022). <https://doi.org/10.1038/s41467-022-33407-5>

[2] Bujotzek, Markus R., et al. "Real-World Federated Learning in Radiology: Hurdles to overcome and Benefits to gain." arXiv preprint arXiv:2405.09409 (2024).

[3] Saldanha, O. L., Quirke, P., West, N. P., James, J. A., Loughrey, M. B., Grabsch, H. I., ... & Kather, J. N. (2022). Swarm learning for decentralized artificial intelligence in cancer histopathology. *Nature medicine*, 28(6), 1232-1239.

[4] Kairouz, P., McMahan, H. B., Avent, B., Bellet, A., Bennis, M., Bhagoji, A. N., ... & Zhao, S. (2021). Advances and open problems in federated learning. *Foundations and trends® in machine learning*, 14(1–2), 1-210.

[5] Sattler, F., Wiedemann, S., Müller, K. R., & Samek, W. (2019, July). Sparse binary compression: Towards distributed deep learning with minimal communication. In *2019 International Joint Conference on Neural Networks (IJCNN)* (pp. 1-8). IEEE.

[6] Karimireddy, Sai Praneeth, et al. "Scaffold: Stochastic controlled averaging for federated learning." *International conference on machine learning*. PMLR, 2020.

[7] Li, T., Sahu, A. K., Zaheer, M., Sanjabi, M., Talwalkar, A., & Smith, V. (2020). Federated optimization in heterogeneous networks. *Proceedings of Machine learning and systems*, 2, 429-450.

[8] Caldas, Sebastian, et al. "Leaf: A benchmark for federated settings." arXiv preprint arXiv:1812.01097 (2018).

(Remarks on code availability)

Version 1:

Reviewer comments:

Reviewer #1

(Remarks to the Author)

I have carefully read through the response from the reviewers, and would like to thank the authors for such effort.

I would say, the authors have resolved all my concerns.

(Remarks on code availability)

Reviewer #2

(Remarks to the Author)

Overall the reviewer considers the authors have answered their main concerns and that the manuscript is in a much better shape now. In particular the reviewer appreciates the effort spent in notably 1. reformulating the abstract to emphasize that FL is simulated as well as 2. highlighting the differences between the tasks and 3. explaining the method to compute "communication efficiency" in details.

However, the reviewer still feels very uncomfortable with the "federated evaluation" wording and would like the authors to change it. As for the exact wording to be used, the authors themselves use "collaborative, multi-site evaluation" I558 and inference in the wild" I1073 which are much better suited and the reviewer is fine with both formulations. In order to emphasize the contribution of this paper on the reproducibility and harmonization of the evaluation protocol, which is significant, the reviewer would accept any expression such as "reproducible multi-site evaluation protocols", expressions including adjectives such as "fair" or even acronyms to facilitate messaging depending on the authors' inclinations as long as they do not include the word "federated".

The reviewer would like to point out that this attention to details on the wording is not simply a reviewer's whim. Federated Learning for medical research is a complex multi-disciplinary field where both transparency and attention to messages and wording is of utmost importance so that medical practitioners and partner institutions' representatives become aware of the challenges and the cost that it currently entails.

Nowadays there is widespread misunderstanding of both the true privacy guarantees offered by FL as well as the difficulty of actually setting up such non-simulated collaborations in practice.

As a part of the community, the reviewer takes some part of responsibility for this current state of things but encourage the authors to participate with the reviewer in the effort to change mentalities by being crystal clear when describing what they are doing.

This article illustrates very nicely such challenges and it is the reviewer's hope that this contribution, alongside like-minded initiatives, will help bring the community closer of true reproducible standardized federated trainings between partners institutions. However, as the paper shows, currently even the multi-site evaluation step and siloed training represents already a huge challenge that the authors tackle without even considering non-simulated federated learning, which is much trickier.

Minor:

- The computation of the "communication efficiency" is now much clearer however it is very mysterious to the reviewer how gaussian sampling helps evaluation or ranking. Could the authors clarify that ?

- Another issue currently present in the article is the lack of details in the Methods especially on each team's contributions. Indeed as asked by the authors in their rebuttal to comment m2.4: it is not blocking but yes, the reviewer would like preferably a more precise description of at least the aggregation methods used by the different teams using mathematical formulas. The reviewer advise the authors to not shy away from throwing a few equations in this very practical article as it would help comprehension a great deal and the reviewer expects readers to gain lots of insights by reading the details of say "FedPIDAvg" or "Adam as a server optimizer" which are too vague. The reviewers understands there are many teams and it is time consuming to do the back and forth but potentially it could be done only for a selection of say the top 3 most innovative aggregation methods.

- Fig 6 is not bringing anything to what Fig 1 already presents. Furthermore it is a reproduction from another article, which is a bit lazy. Considering Figure 5, The reviewer would like the U-net structure to be put in perspective using the radiology segmentation task that is studied here at least and potentially integrated with Fig 6 if necessary to emphasize the federation part (although Figure 1 is already quite clear). Maybe emphasize the specificities of nn-Unet and the fact that most participants chose the same architecture ? As a recap, neither Figure 5 nor Figure 6 seem to bring anything to the article in their current state. The reviewer would like them to be either improved/replaced by better figures from the supplementary or removed.

- Even though the reviewer likes the new wording used to compare this work with clinical trials. The reviewer would like that comparison removed entirely as the current evaluation protocol is nowhere near achieving the standards of clinical trials.

(Remarks on code availability)

From a very very quick pass through the repository <https://github.com/FETS-AI/Challenge>.

Positive points:

- There are lots of detailed instructions, READMEs, documentations and links to websites to make a submission
- it uses docker therefore implies some reproducibility but through mlcube, which is an abstraction the reviewer is not familiar with

- Q&A forums accessible

- Tasks 1 and 2 correspond to two different distinct folders helping readability

Negative Points

- no tutorial / quickstart clearly accessible from the landing README

- not lightweight/self-contained at all as it uses a mix of R, python code as well as multiple (and somewhat intricate)

dependencies

- linked to above points but it seems quite involved to just run some code and will make reproduction of the results quite the endeavor

The reviewer is not sure that the claims in the code reporting summary are supported by the code notably the availability of a small lightweight example to quickly test the code as well as the precise timings on standard hardware but the reviewer did only a very quick pass and might be mistaken.

Version 2:

Reviewer comments:

Reviewer #2

(Remarks to the Author)

The reviewer thank the authors for all this rebuttal work which really improved the paper: the flow is much more straightforward and to the point and the contributions, which are significant, really stand out much better now that the wording has been changed. The reviewer is happy with the current state of the manuscript and is hoping to see it published.

(Remarks on code availability)

Response to the reviewers

The authors would like to thank the reviewers for their constructive comments and suggestions that have helped improve the quality of this manuscript.

We thank the reviewer for the constructive and insightful feedback.

The manuscript has undergone a thorough revision according to the reviewers' comments. Please see our responses below, where we also describe how we revised the manuscript based on the reviewer's feedback.

For the reviewers' convenience, we have highlighted significant changes in the PDF with the revised manuscript in red.

Reviewer 1

Reviewer #1 (Remarks to the Author): Expert in AI in healthcare, computer vision, and computational radiology

This work is of significant merit as it is “the very first to establish a decentralized algorithmic benchmarking paradigm for both federated learning (FL) and federated evaluation (FE)”. It is particularly noteworthy for constructing a large-scale FE benchmark incorporating data from 32 globally distributed sites and benchmarking 41 models. Considerable effort has been dedicated to developing this innovative FE benchmark, which could serve as a foundational step towards creating generalizable algorithms for cross-center applications. Therefore, the significance of this work certainly merits recognition.

I have the following concerns:

1. In the FE task, only 5 out of 35 registered teams submit results for this task. It is regrettable that the few submissions (only 5) did not yield many intriguing conclusions, and models from BraTS that were not specifically optimized for FE testing still ranked at the top. What's the main reason for the little engagement? What can be done to improve the submission rate?

RESPONSE:

The low engagement in task 2 (FE) surprised us, too, given that it used the same imaging data as the popular BraTS 2021 challenge and offered additional methodological research opportunities with the provided institution partitioning. While we cannot be sure what are the main reasons, we suspect some of the following played a role. We also provide potential improvements for the future after each point:

- Uncommon setup of a federated challenge: The FeTS challenge pioneered federated (multicentric) evaluation in biomedical challenges, so this setup is rather uncommon. As reviewer 2 pointed out, federated evaluation is not even a standard term in the community and may be associated more with federated learning than cross-site (domain) generalization. Thus, researchers interested in this topic might have missed that task 2 of the FeTS challenge was not about federated learning. This hypothesis is supported by email communications initiated by M. Zenk after the FeTS 2021 challenge: we asked external researchers with works on domain generalization for medical image segmentation whether they had been aware of the FeTS challenge and why they did not participate. In all 6 responses we received, the researchers said they had not been aware of the challenge and they expressed their interest in the challenge. As this was personal communication via email, the responses could be more positive than they would be in an anonymous survey, but they are an indication that the basic concept is interesting and the

main issue was making people aware of the initiative. However, even our increased advertising efforts in 2022 seem to have been insufficient, as they only increased the number of participants slightly from 3 to 5. Compared to this, the BraTS 2022 challenge attracted 10 participants, which could have submitted to the FeTS challenge 2022, too, with minor extra effort.

- *Potential improvements:* Future challenges with multi-site generalization evaluation should not be combined with a (simulated) federated learning task and instead they should stress the generalization aspect in their title. If they are designed as a large-scale “phase-2 challenge”, their relation to a previous “phase-1” (BraTS 2021 in our case) challenge should be stressed.
- Training data novelty and imbalance: Many participants join a challenge because of the novel datasets published with it. In the FeTS challenge, we re-used the BraTS challenge training data and added institution partitioning files, which assign each case to a different institution. The training institution partitioning was strongly imbalanced in FeTS (cf. Fig. A1), which is realistic in a multicentric setup but may also have been a methodological hurdle for participants. Hence, it could be that the novel meta-information in the FeTS challenge was not attractive enough for potential participants, or the short timeframe of the challenge imposed by the MICCAI conference did not allow innovative methodological work. This point is also supported by the reduced participation in the BraTS 2022 challenge (10 teams) compared to the BraTS 2021 challenge (61 teams), which is likely due to lack of additional data in 2022.

Potential improvements: For similar challenges in the future, it could be beneficial to further extend the training dataset (if there had been a predecessor challenge), provide more detailed meta-data for each case (e.g., not just which institution ID but also information on the patient and scanner) or collect data from different sites (domains) in a more balanced fashion.

- Although only 5 official submissions were made to Taks 2, we would like to stress that the 36 other models evaluated in the analysis represented the state of the art in brain tumor segmentation. They were trained on the same multi-centric images, even though they were unaware of the center-split. Previous works like the M&Ms challenge have shown that such simple “pooled” training strategies can outperform more specialized methods, so the FeTS challenge still provides interesting insights into the real-world applicability of automatic segmentation algorithms.

We added a paragraph in the discussion section (lines 876 ff.) on the two possible issues and solutions described above.

2. In the FE task, the evaluation currently requires accessing data stored on each site’s server. How can this benchmark be maintained in the long term after the challenge ends, so that the following research can continue and benefit from it? Alternatively, how much test data can be accessible to the community after overcoming privacy concerns?

RESPONSE:

This is indeed an important point. In challenges without federated test data, organizers sometimes publish their test data (e.g. M&Ms [R1.2.1]) or provide a continuous evaluation online platform (e.g. BTCV [R1.2.2] or BraTS [<https://www.synapse.org/Synapse:syn53708126/wiki/626320>]). For federated evaluation challenges a continuous evaluation platform could also be an option if the testing data can be shared with the organizers after the challenge. In the FeTS challenge, this was only possible for about half of the test cases. Often federated data cannot be shared, so federated evaluation workflows need to be repeated (e.g. annually). This was not planned for the FeTS challenge, but once the federated infrastructure is set up, future evaluations are expected to require considerably less effort. Changes in staff, hardware or software at individual sites are still potential hurdles for maintaining a federated evaluation benchmark. Benchmarking initiatives like MLCommons could play a role in the technical maintenance of such federated efforts, which is why we continue to collaborate with them and the MedPerf project in particular.

A short paragraph on this topic was added to the discussion section of the manuscript (Lines 871-875).

[R1.2.1] Campello, Victor M., et al. "Multi-centre, multi-vendor and multi-disease cardiac segmentation: the M&Ms challenge." IEEE Transactions on Medical Imaging 40.12 (2021): 3543-3554.

[R1.2.2] Landman, B., Xu, Z., Igelsias, J.E., Styner, M., Langerak, T., Klein, A.: Miccai multi-atlas labeling beyond the cranial vault–workshop and challenge (2015), <https://www.synapse.org/#!Synapse:syn3193805/wiki/217760>

3. In the FL task, introducing communication efficiency in evaluation is reasonable. However, the details of normalization on hardware systems, as mentioned in line 921 “To calculate this metric, we simulate the time taken to run each round, and it is normalized across a single hardware system, so that final scoring will not depend on the hardware used.”, were not clarified.

RESPONSE:

We thank the reviewer for making us aware that this part of the evaluation methodology is unclear. Below, we provide an accurate definition of the metric and how the FL round times are simulated in the experiments. These are added to methods sections 4.2 and 4.4.2 of the revised manuscript, respectively.

Though we used the term ‘communication efficiency’ metric, the intention was to measure how quickly the method reached good DSC scores, with ‘quickly’ measured in terms of simulated wall clock time as discussed below. The use of the term ‘communication efficiency’ can be understood loosely in terms of the ability for methods with higher scores in this metric to stop the federation early and still have a high performing model. Stopping early would reduce the amount of information communicated in the training run, and therefore increase the efficiency of communication. We agree that this was a misleading term, so we rename it to ‘convergence score’ and modified the text to clarify the intent.

- Definition of the convergence score metric: In each round of an FL experiment, the mean validation DSC and simulated round time T (details on how it is simulated below) is computed for each FL client. Over the course of an experiment, this results in a DSC-over-time curve. The validation DSC can in some cases decrease at later times (e.g. due to overfitting or randomness in the optimization), but as the model with best validation DSC is used as the final model such a decrease should not be penalized. Therefore, a “projected” DSC curve is computed by:

$$DSC_{\text{proj}}(t) = \max_{t' < t} DSC(t')$$

The “convergence score” metric is calculated as the area under the projected-DSC-over-time curve.

- Realistic federated learning timings: To simulate the FL training time, we used a subset of the real-world times measured in a previous study [R1.3.1]. Note that the simulated time is different from the program runtime; it is rather an estimate of the wall time such an FL experiment would take in a real federation similar to [R1.3.1]. Specifically, the simulated time is comprised of four components: training time, validation time, model weight download and upload time. In each round, the simulated time for each collaborator k is

$$T_k = T_{\text{down},k} + T_{\text{up},k} + T_{\text{val},k} \cdot N_{\text{val},k} + T_{\text{train},k} \cdot N_{\text{train},k}$$

and the total time for each round is the maximum of T_k across collaborators. Each of the timing terms in above equation is obtained by sampling from a normal distribution. The mean and standard deviation are fixed for each collaborator and simulated time component. They are selected randomly from the time measurements performed during the FeTS initiative [R1.3.1], and can be found in [our code](https://github.com/FeTS-AI/Challenge/blob/2c3e2cc01d23af936567ad6c602dda9dc2f87192/Task_1/fets_challenge/experiment.py#L38) (https://github.com/FeTS-AI/Challenge/blob/2c3e2cc01d23af936567ad6c602dda9dc2f87192/Task_1/fets_challenge/experiment.py#L38)

- Hardware used for final training: All task-1 submissions were retrained to guarantee a fair competition. The retraining runs were distributed among several machines to speed up the process. As the simulated run times were generated with a fixed random seed, they were independent of the hardware and in that sense “normalized”.

Revisiting the convergence score metric also made us aware that it was used incorrectly for computing the ranking score for Task 1, treating it as a metric for which lower values are better. The revised manuscript corrected this. Although this only affects Task 1 and does not change the contribution of this challenge, changes in the results section 2.2 were required.

[R1.3.1] Pati, Sarthak, et al. "Federated learning enables big data for rare cancer boundary detection." *Nature communications* 13.1 (2022): 7346.

4. In the FL task, can you provide more details on how the organizers simulated federated data based on the training data of BraTS21?

RESPONSE:

The federation was simulated with the OpenFL tool [R1.4.1], which creates federated “clients” on a single host machine. The data distribution among those FL clients followed a natural partitioning by institution. This was possible because the training data (originally published in BraTS21) was acquired at 23 institutions.

We retrained all submitted algorithms to ensure that no data leakage between federated sites occurred, which is crucial for a fair comparison of FL algorithms. The FL simulation was limited to one week of simulated FL time (see our response to this reviewer’s comment 1.3 for how it was simulated), which is the same limit also used by the participants during method development and a realistic time horizon to use in real-world implementations such as the FeTS initiative [R1.4.2]. The dataset partitioning for retraining (partitioning 3) was a refinement of the geographic partitioning (partitioning 1), which split the largest institution into 7 parts based on additional available meta-data for this institution. Specifically, this institution comprises a system of hospitals in close geographical proximity, which were combined for partitioning 1 and re-grouped into 7 parts for partitioning 3. The reasoning behind using an unseen partitioning for retraining was that the participants should develop generalizable FL algorithms that do not overfit on a particular client partitioning.

Sections 4.1.4, 4.4.2 and 4.4.3 were updated with the details we described here.

[R1.4.1] Foley, Patrick, et al. "OpenFL: the open federated learning library." *Physics in Medicine & Biology* 67.21 (2022): 214001.

[R1.4.2] Pati, Sarthak, et al. "Federated learning enables big data for rare cancer boundary detection." *Nature communications* 13.1 (2022): 7346.

5. In the FE task, information on data distribution of different sites (e.g., age, ethnicity, vendor/manufacture) is lacking, which would facilitate understanding the challenges of generalizability in this benchmark, and potentially help the development of algorithms with better generalizability.

RESPONSE:

Quantifying the test data diversity was possible for our study using metadata shared by the federated institutions. Sharing individual patient’s data was not possible in the federation, so we only have summarizing statistical information for: sex (% male/female/unknown), IDH status (% wild type/mutated/unknown), age (mean, std, min, max). Figure A2 of the revised manuscript visualizes the meta-data distributions. Especially in the IDH and age statistics, there notable differences that illustrate the diversity. Information about the patients’ ethnicity was not available, but geographical diversity could be used

as a proxy. Fig. 1 in the manuscript shows the institutions on a map; the exact counts per continent are: North America (13), Europe (9), South America (4), Asia (3), Africa (1), Australia (1).

Apart from patient-specific data, we also obtained information about the scanners and procedures used for image acquisition from 20 of 32 sites. In total there were 4 different vendors (Siemens, GE, Philips, Hitachi) and some institutions used more than one scanner vendor/model. The rough count of unique model names (only considering differences in the first part of the model name, e.g. Siemens *Aera*) are: Siemens (12), GE (4), Philips (3), Hitachi (1). Furthermore, different acquisition planes (axial/sagittal), field strengths (1.5T/3T) and MRI coils were used. In conclusion, the diversity in terms of image acquisition is large in the federated test data used for Task 2.

We added the description above and the figure to Appendix A of the supplementary material.

6. Considering the federated data is simulated from the BraTS21 dataset, it would be interesting to assess and analyze how the performance of segmentation algorithms developed under FL compares to directly using the BraTS21 dataset. In other words, to what extent can current FL algorithms approach this baseline? This may shed light on the potential and meaning of this FL challenge.

RESPONSE:

The centrally trained baseline is often found in FL papers as an upper baseline, as FL training usually cannot outperform a model trained on the same, pooled data. We thank the reviewer for this suggestion and add the baseline to our results. Additionally, a lower baseline interesting for this challenge is the “default FL” algorithm provided by the challenge organizers as a template.

For the centralized baseline, we only tuned the learning rate (and its schedule), because the participants were restricted in the same way. However, we did not impose limits on the maximum training time, because such limitations would be unrealistic for centralized training. The validation DSC plateaued in the pooled training after roughly 10 epochs, which is similar to the number of FL rounds feasible within the maximum simulated time with full client participation and one epoch per round.

The default FL baseline shows how much the teams could improve the performance during the challenge. It uses FedAvg with the default hyperparameters in the template provided to the challenge participations and selects all sites for training in each FL round.

The results for the centralized baseline (table B1 added to the revised manuscript and inserted below) show that the gap between FL and pooled training is not closed for the FeTS challenge setting. Note that the scores here are not comparable to the Task 2 submissions, because different test sets are used and also because Task 2 participants had complete freedom in all algorithmic design choices, while Task 1 participants all used the same segmentation model and local training.

The “Default” FedAvg baseline performs worse than the top teams, especially for the Hausdorff and convergence score metrics, highlighting that client selection and weight aggregation methods can improve performance and efficiency. A comparison with the results of the teams with worse ranking, however, also shows that FedAvg with full participation is still a strong baseline in cross-silo settings, which can outperform more complicated methods in a subset of metrics.

Table B1: Overview of metric values for each Task 1 team, where \uparrow/\downarrow means higher/lower values are better. The mean across 570 test cases is shown here, except for the convergence score, which is a single number per team. Color maps are applied per metric column, ranging from dark red (worst) to dark green (best). Note that the ranking score is not computed from the mean metric values shown here but by case-based ranking as described in section 4.4.3. The ‘Default’ and ‘Centralized’ models are not ranked, as they are baselines and not part of the competition.

Team	DSC \uparrow			HD95 \downarrow			Conv. score \uparrow	Rank score \downarrow
	WT	TC	ET	WT	TC	ET		
FLSTAR	75.8	75.1	72.8	31.3	29.9	29.1	72.7	2.75
Sanctuary	76.7	76.3	73.5	24.5	32.6	32.5	71.3	3.05
RoFL	77.5	77.8	74.7	28.9	28.8	29.6	70.2	3.35
gauravsingh	72.4	68.6	66.3	25.2	32.7	32.4	71.5	3.67
rigg	76.8	76.9	74.2	24.4	32.5	32.8	30.1	4.65
HTTUAS	72.5	71.7	67.5	35.9	33.8	35.2	69.0	4.69
Flair	50.5	44.4	50.1	28.6	51.3	44.4	41.8	5.85
Default	74.8	76.2	73.3	37.3	34.0	34.8	69.3	-
Centralized	81.8	80.5	77.2	19.1	31.7	31.3	-	-

These results were added to the revised manuscript in the supplementary material B.

7. The number of image scans across different sites is quite imbalanced (Fig. C8), as well as their geographical regions. Although it shows significant improvement in size and diversity compared to the BraTS dataset, and this imbalance is taken into account in rankings computation, it would be beneficial to alleviate this imbalance through further data collection.

RESPONSE:

We agree that future work should focus on extending the data especially for currently underrepresented patient populations, e.g. from certain geographical regions. This should come hand in hand with more meta-data collection to quantify distribution shifts and analyze potential reasons for the lack of generalization. This point was added to the manuscript discussion as a future direction.

8. A small typo: The reference to section 2.4 in line 1724 of the supplementary materials seems incorrect; perhaps it was intended to refer to section 4.1.3?

RESPONSE:

Thank you for pointing out this mistake. We added references to sections 4.1.3 (annotation protocol) and 2.5 (results of the quality control).

Reviewer #1 (Remarks on code availability):

The code is in good shape.

RESPONSE:

The authors would like to thank the reviewer for appreciating the provided code.

Reviewer 2

Reviewer #2 (Remarks to the Author): Expert in federated learning, AI in healthcare and oncology, and computational pathology

Recap:

The reviewer finds this study of great interest as it 1. contains extensive evaluations of models on real-world healthcare data from numerous institutions 2. depicts the practical hurdles associated with doing multi-sites ML with medical institutions, notably very pragmatic considerations such as coordinating partners (e-mails, zoom calls, etc.), IT-setup (reproducibility, hybrid IT servers, etc.), data quality checks wo cheating, which complements nicely findings already reported by [1, 2], 3. inspects closely failure cases of models, which goes further than quantitative metrics such as DICE and are of utmost importance for medical research. It is the belief of the reviewer that ultimately such findings should be published.

However the reviewer finds that there are a few major (and minor) points on which the authors should provide detailed answers before this manuscript can be considered ready for publication.

The reviewer lists those points below.

Note that the reviewer only commented on points related to their domain of expertise.

Major:

1. Although Figure 1 indeed labels federated learning as operating on "simulated federated data", the fact that the federated learning part is only simulated is not put forward enough in the paper. This is a major concern for the reviewer as the reviewer feels some sentences are a bit misleading such as the authors writing in the abstract that their work stand out from related initiatives as "[other related works only] simulat[e] real-world conditions in a controlled environment" l489, which is exactly what the authors themselves are doing for the federated training part. The reviewer emphasize that they do not consider simulating FL as a weakness of the paper as long as it is clear in the paper see i.e. [3], which simulates FL but is interesting nonetheless. Readers even have to wait until Methods to understand "that training on pooled data was allowed in this task" l985 so authors are comparing pooled methods with no restrictions and federated methods with potentially restrictions on the number of rounds or on the "privacy" of communicate updates but even those restrictions are not clear see point 5. The authors' non simulated contribution exists but is on the evaluation part.

RESPONSE:

Two concerns are expressed by the reviewer in this comment:

- Unclear language about the simulation of FL in task 1: We thank the reviewer for this feedback and revisited the manuscript text for formulations which might be misleading in that regard. Specifically, l. 489 referred to the federated evaluation task, but we rephrased the sentence to avoid the impression that we perform non-simulated FL. Further modifications were made in the following places, which should prevent any confusion:
 - Abstract: We clarified that FL simulations are performed while FE is truly distributed.
 - Introduction: We removed ambiguity in our objectives and contributions.
 - Methods: We clearly state in the first sentence of section 4.4.2 that FL is simulated.
 - Results: We clearly state in each subsection which Task it refers to and also mention for Task 1 that it simulates FL.
- Comparing pooled and federated methods (see this reviewer's point 2.5): We think that this concern is based on a misunderstanding and we would like to stress that training on pooled data was only allowed for Task 2. The

submissions for both tasks are independent, so Task 1 only compared FL methods, while the submissions to Task 2 all used pooled training. Apparently, our writing lacked clarity in this regard, so we improved the text accordingly. More details on this in our response to point 2.5.

2. The claim "the 1st FL challenge conducted" I564 is a bit vague and therefore the reviewer is not sure if it is valid for instance was LEAF in 2018 a challenge [6] ? What is the precise definition of a challenge apart from "Computational competitions" I1 ? Is it to have a private leaderboard ? The reviewer is not sure even with such definition that this work is the first of its kind. The weaker claim that authors do later "[authors are] the first to circulate the solutions provided by challenge participants across multiple collaborating healthcare sites" I595 within a challenge is already strong but truer: it might be true within the context of a challenge but external validation in other medical institutions is standard for ML applications in healthcare and has been widely used by most works published in nature-type journals. However the reviewer agrees that the way the authors do it is novel (providing docker images + QC instructions).

RESPONSE:

This comment contains questions on two claims made in the manuscript. We provide our perspective on each point below and also describe how we clarified the claims in the manuscript.

1. Task 1 represents the very first FL challenge.

A possible challenge definition is formulated in L. Maier-Hein et al. (2018): "Challenge: open competition on a dedicated scientific problem in the field of biomedical image analysis. A challenge is typically organized by a consortium that issues a dedicated call for participation. A challenge may deal with multiple different tasks for which separate assessment results are provided." This definition leaves several details open, so we offer our interpretation and reasoning for calling FeTS the first challenge that evaluates FL methods below.

Apart from the embedding in the MICCAI conference and the resulting condensed timeline and increased publicity, the main difference to public benchmarks such as LEAF is that the test set is (as the reviewer points out) not public for challenges, which guarantees a fair comparison for continuous evaluation. In the case of FL challenges, there is another important aspect of performing a fair evaluation: Making sure that all participants implement FL correctly, in particular avoiding (accidental) data leakage, and reproducible measurement of communication efficiency. In the FeTS Challenge Task 1, we provided a suitable software framework and retrained all algorithms on standardized hardware, to fulfill both aspects. In this sense, to the best of our knowledge, Task 1 of the FeTS challenge 2021 was indeed the first challenge that benchmarked FL algorithms. We are aware of one other FL challenge conducted so far [R2.2.0], which took place after the FeTS challenge 2021 and at the same time as FeTS 2022.

We made the claim in the abstract more precise and added a paragraph to the introduction that clarifies the relation to non-challenge FL benchmarks like LEAF, and also highlights the benefits of a challenge setting.

2. The federated evaluation performed for Task 2 was the first to circulate participant submissions across multiple sites. As the reviewer said, we are not the first to perform a federated evaluation in general. In fact, all "real-world" FL studies also perform federated evaluation, for example the FeTS initiative [R2.2.1], but also [R2.2.2] and [R2.2.3]. Our claim only refers to the competition/challenge setting, which faces unique difficulties, like a diversity of algorithms and software implementations.

The reviewer also mentions that studies with external (non-federated) validation on data from other institutions are

common nowadays, due to their importance for practical applicability. Indeed, the main motivation for Task 2 is to scale this validation strategy up and introduce it to the challenge setting. As we note in the manuscript, challenges like the M&Ms challenge previously performed validation on data from unseen institutions, but the scale and the use of federated evaluation distinguishes the FeTS challenge from previous efforts. We hope that a FE within the challenge setting (as shown during the FeTS challenge) would enable a paradigm for thorough generalizability evaluation of methods developed by the challenge participants for the specific /focused task of an associated challenge.

We made the claim in the abstract more precise and added a paragraph to the introduction mentioning that other groups have performed federated multi-centric evaluation before, but not in a challenge setting.

In conclusion, we intended to make these claims only restricted to the challenge setting, for which we think both are valid. The introduction was revised to clarify our motivation and potential benefits of studying FL and cross-site generalization in a (federated) challenge setting.

[R2.2.0] Schmidt, Kendall, et al. "Fair evaluation of federated learning algorithms for automated breast density classification: The results of the 2022 ACR-NCI-NVIDIA federated learning challenge." *Medical Image Analysis* 95 (2024): 103206.

[R2.2.1] Pati, Sarthak, et al. "Federated learning enables big data for rare cancer boundary detection." *Nature communications* 13.1 (2022): 7346.

[R2.2.2] Ogier du Terrail, Jean, et al. "Federated learning for predicting histological response to neoadjuvant chemotherapy in triple-negative breast cancer." *Nature medicine* 29.1 (2023): 135-146.

[R2.2.3] Dayan, Ittai, et al. "Federated learning for predicting clinical outcomes in patients with COVID-19." *Nature medicine* 27.10 (2021): 1735-1743.

3. Related to point 2. the term "federated evaluation" which the authors use multiple times is not well-defined nor standard in the community (the 2 references the authors provide using the term "federated evaluation" 77 and 115 in the paper are a somehow cherry-picked and have very few citations). The reviewer understands that authors use that term to describe the reproducible docker image that authors send to different medical institutions and the associated testing protocol, which are very interesting by themselves but 1. it should be clearly defined in the text what it means and 2. the use of the "federated" adjective is in the opinion of the reviewer misleading. The reviewer strongly encourages authors to use clearer naming such as "reproducible multi-sites evaluation protocol" or something along those lines as the reviewer understands for instance as non FL methods are tested (see 1985) "federated evaluation" is not i.e. the evaluation of federated learning methods.

RESPONSE:

Although "federated evaluation" (FE) is not a standard term, we are not the first to use this term (e.g. [R2.3.1], and [R2.3.2], [R2.3.3] in the context of model personalization). However, there are also some papers that do not call it FE (e.g. [R2.3.4]). We agree that FE should be precisely defined if used. The "Task 2" box in figure 1 already captures what we mean with federated evaluation, but also added a definition in the introduction of the revised manuscript, as the collaborative, multi-site evaluation of (segmentation) models.

The main concern expressed by the reviewer appears to be potential confusion whether FE implies that the evaluated models were trained with FL, as FE is not a standard term. With the definition of FE from the introduction and the added remark that centralized training was allowed for Task 2, we think that this particular risk is low. Furthermore, we think that

the term “federated evaluation” captures the essence of the evaluation procedure, as it implies the evaluation is both distributed (multi-site) and collaborative. In fact, other “federated analytics” workflows are conceivable, as described by [R2.3.5].

We would like to ask the reviewer if our definition and clarifications above appear sufficient to keep the name “federated evaluation” or if another term is still preferred by the reviewer.

[R2.3.1] Karargyris, Alexandros, et al. "Federated benchmarking of medical artificial intelligence with MedPerf." *Nature machine intelligence* 5.7 (2023): 799-810.

[R2.3.2] Paulik, Matthias, et al. "Federated evaluation and tuning for on-device personalization: System design & applications." *arXiv preprint arXiv:2102.08503* (2021).

[R2.3.3] Wang, Kangkang, et al. "Federated evaluation of on-device personalization." *arXiv preprint arXiv:1910.10252* (2019).

[R2.3.4] Dou, Qi, et al. "Federated deep learning for detecting COVID-19 lung abnormalities in CT: a privacy-preserving multinational validation study." *NPJ digital medicine* 4.1 (2021): 60.

[R2.3.5] Elkordy, Ahmed Roushdy, et al. "Federated analytics: A survey." *APSIPA Transactions on Signal and Information Processing* 12.1 (2023).

4. The term "communication costs" is also used frequently by the authors but in a sense that is not clear and potentially different to what is meant in most federated learning literature [4]. In the literature this term is used to describe the total amount of data that is sent by participants to the server or the number of rounds of FL see i.e. [5] which proposes clever sparsification and compression mechanisms on the FL updates. What the authors mean in this work is unclear. Searching for "communication" gives few hits the most relevant one seems line 921 "Communication efficiency is an additional metric used for Task 1. To calculate this metric, we simulate the time taken to run each round, and it is normalized across a single hardware system, so that final scoring will not depend on the hardware used" and 1926 "The communication efficiency metric will be computed as the area under the validation learning curve over 1 week of simulated time". While the reviewer agrees there might be a light coupling between the number of parameters sent and this normalized simulated time. How is this computed when methods are pooled ? Is there a penalty being applied ? Can the authors provide a more thorough definition of this metric and how it is computed for FL methods vs pooled methods ? The reviewer understands sampling client plays a role in this ranking but is it not clear to the reviewer why ? Are training runs' times in the different institutions counted sequentially ? Is round time dominated by one participant having lots of training data ? Notably authors seem to also take into account the cost of communication where communication in this case refers to sending e-mails and doing zoom calls "which were fixed with remote support on shared log files, emails and video calls. This resulted in slow feedback loops and revealed communication as a primary bottleneck" 1722. Could the authors clarify this point with equations potentially or detailed protocol with an example ?

RESPONSE:

We thank the reviewer for highlighting this issue. Incorporating typical constraints of FL projects like limited communication into a benchmark is not trivial. While the total amount of data sent is one option, we chose a different way to introduce realistic FL constraints and measure efficiency. Both reviewers agree that our description of the communication efficiency metric lacks important details. We follow their recommendation and provide a precise description in the methods sections 4.2 and 4.4.2 of the revised manuscript. Here, we first describe what we did and provide our reasoning behind it. At the end, we also discuss the benefits and drawbacks of our approach.

Though we used the term ‘communication efficiency/cost’, the intention behind the metric was to measure how quickly the method reached good DSC scores, with ‘quickly’ measured in terms of simulated wall clock time as discussed below.

The use of the term ‘communication efficiency’ can be understood loosely in terms of the ability for methods with higher scores in this metric to stop the federation early and still have a high performing model. Stopping early would reduce the amount of information communicated in the training run, and therefore increase the efficiency of communication. We agree that this was a misleading term, so we rename the metric to ‘convergence score’ and modified the text to clarify the intent.

- Definition of the convergence score metric: In each round of an FL experiment, the mean validation DSC and simulated round time T (details on how it is simulated below) is computed for each FL client. Over the course of an experiment, this results in a DSC-over-time curve. The validation DSC can in some cases decrease at later times (e.g. due to overfitting or randomness in the optimization), but as the model with best validation DSC is used as the final model such a decrease should not be penalized. Therefore, a “projected” DSC curve is computed by:

$$DSC_{\text{proj}}(t) = \max_{t' < t} DSC(t')$$

The “convergence score” metric is calculated as the area under the projected-DSC-over-time curve over one week of simulated time.

- Realistic federated learning timings: To simulate the FL training time, we used a subset of the real-world times measured in a previous study [R1.3.1]. Note that the simulated time is different from the program runtime; it is rather an estimate of the wall time such an FL experiment would take in a real federation similar to [R1.3.1]. Specifically, the simulated time is comprised of four components: training time, validation time, model weight download and upload time. In each round, the simulated time for each collaborator k is

$$T_k = T_{\text{down},k} + T_{\text{up},k} + T_{\text{val},k} \cdot N_{\text{val},k} + T_{\text{train},k} \cdot N_{\text{train},k}$$

and the total time for each round is the maximum of T_k across collaborators. Each of the timing terms in above equation is obtained by sampling from a normal distribution. The mean and standard deviation are fixed for each collaborator and simulated time component. They are selected randomly from the time measurements performed during the FeTS initiative [R1.3.1], and can be found in [our code](https://github.com/FeTS-AI/Challenge/blob/2c3e2cc01d23af936567ad6c602dda9dc2f87192/Task_1/fets_challenge/experiment.py#L38) (https://github.com/FeTS-AI/Challenge/blob/2c3e2cc01d23af936567ad6c602dda9dc2f87192/Task_1/fets_challenge/experiment.py#L38).

Based on this definition, we can answer two questions by the reviewer:

- “The reviewer understands sampling client plays a role in this ranking but is it not clear to the reviewer why?” Client sampling can affect the area under the (t-DSC) curve, because the simulated time for one FL round can be reduced by training only on a subset of clients as mentioned above. By choosing the clients that improve the validation DSC, which is always computed across all clients, score more quickly, the training also becomes more efficient and the AUC increases in terms of reaching greater DSC sooner, which in turn increases the AUC of DSC over simulated time.
- “Are training runs’ times in the different institutions counted sequentially? Is round time dominated by one participant having lots of training data?”

The total simulated time for one round is the maximum of the runtime of all clients that participated in the last round. As the training time increases with number of samples, the client with the largest dataset also has the largest impact on the round time, as the reviewer suspected and how it would also be in reality. The correlation of sample size and round time is illustrated in figure B7, which we added to the supplement of the revised manuscript.

We replaced “communication cost/efficiency” in the manuscript with formulations that focus on the overall efficiency of the FL algorithm, from the perspective of the convergence score metric. As this metric has limitations and may not be the perfect choice for other FL applications, we added a paragraph in the discussion (lines 822 ff.) on this topic.

Revisiting the convergence score metric also made us aware that it was used incorrectly for computing the ranking score for Task 1, treating it as a metric for which lower values are better. The revised manuscript corrected this. Although this only affects Task 1 and does not change the contribution of this challenge, changes in the results section 2.2 were required. Two questions by the reviewer were not addressed above, because we think they are based on the misunderstanding from reviewer comment 5 (below):

- “How is this computed when methods are pooled?”

We do not compute this metric for methods that train on the pooled data (i.e. models submitted to Task 2). More on this point in our response to R2.5 (below).

- “Notably authors seem to also take into account the cost of communication where communication in this case refers to sending e-mails and doing zoom calls [...] 1722. Could the authors clarify this point with equations potentially or detailed protocol with an example?”

We regret the confusion between our ‘communication efficiency’ metric and the comments we made about communication during the competition in the form of emails and video calls. This section of the manuscript refers to the organization of the federated evaluation (Task 2). To organize this federated evaluation study, communication between the organizers and evaluation sites was necessary. These efforts were made in the challenge preparation and execution (e.g. to resolve technical issues). Our comments here were intended to highlight the work involved in coordinating a competition with federated evaluation. To avoid confusing readers, we clearly stated at the beginning of each results section which challenge task it refers to.

5. One of the weakest part of this paper is indeed the evaluation setup and the comparison between participating teams. It seems there are very few constraints on what participants can do apart from the ones listed 1961 to 971 and therefore it is difficult for the reviewer to assess if the different approaches all operate on an equal footing. The reviewer thinks some information is missing such as: Is the number of rounds fixed ? In this case how can the pooled method be valid ? How can the test-time batch normalization methods be valid ? While a setting with few constraints might be warranted, submissions should be clustered and compared between roughly similar algorithms and training/testing protocols. i.e. say model ID p

to q use true FL, model X to Y use pooled data, model ID X' use cyclic averaging, model ID X'' use test-time batch-normalization adaptation etc. Could the authors provide a results table with such categories ?

RESPONSE:

We thank the reviewer for this comment, as it might indicate a misunderstanding that can be avoided by improving our presentation. We emphasize that we *do not* compare results of FL methods with non-FL methods in this study:

- Task 1 only compares FL methods (results for the segmentation and communication metrics are in figures B3-5 of the revised manuscript). Constraints: The participants were restricted in their method development by the constraints in l.961-971 of the original manuscript (section 4.4.2). Furthermore, the total *simulated* FL training time was restricted to one week, which was chosen based on the experiences from the real-world FL study in the FeTS initiative [R2.5.1]. This limit indirectly also affects the possible number of FL rounds. The experiments were terminated once the simulated time exceeded the maximum allowed time and the model with highest validation score before the last round was used as the final model, to make sure that a long last round exceeding the time limit does not benefit the participant.
- Task 2 has no methodological constraints, so both FL and centralized methods would be allowed. Although FL-based models were allowed for task 2, too, no Task-2 submission used FL, probably as it cannot compete against centralized training. Therefore, figure 2 only compares methods with pooled training. Already before the challenge started, we decided to allow test-time adaptation (without access to ground truth of course), in order to make the challenge methodologically more interesting. However, to ensure a fair comparison, all participants received the same inference time limit of 180s per case, which was measured and ensured before the federated evaluation on a single reference machine. We did not include submissions to Task 1 in the federated evaluation, because of the tight challenge schedule: the evaluations needed to be finished before the MICCAI conference and for Task 1, retraining all submissions (which was necessary to prevent cheating) already took a lot of time.

A sentence was added in the introduction to clarify that pooled training was only allowed for Task 2 and we also clarified this in the results section 2.3.

We thank the reviewer for the constructive feedback, suggesting a table with an overview of algorithm categories/characteristics. Such an overview can be found in the revised manuscript for both tasks:

- For Task 1, we summarized the essential algorithm characteristics in the table 1 of the revised manuscript. Additionally, the detailed participants' method description was improved, as requested in the minor comment of this reviewer, and moved to the methods section. Sections 2.1 and 2.2 are adapted to incorporate these characteristics.

Table 1: Algorithm characteristics (aggregation method, learning rate (LR) schedule and client selection) and mean ranking score (column "Score") for all teams that submitted algorithms to Task 1. Algorithms are listed in the order of ranking, with the best on top. The method for computing the ranking score is described in section 4.4.3. A common pattern for aggregation methods is to combine multiple normalized weight terms either through an arithmetic mean or a multiplicative averaging. Selectively sampling clients was used by many teams to improve the convergence speed.

Team	Aggregation Method		LR Schedule	Client Selection	Score
	Weight terms	Combination			
FLSTAR	dataset size val. loss	multiplicative avg.	constant	6 largest	2.75
Sanctuary	dataset size parameter distance validation DSC	multiplicative avg.	polynomial	alternating: all, drop slow clients	3.05
RoFL	dataset size parameter distance	multiplicative avg.; server optimizer	step	all	3.35
gauravsingh	dataset size val. loss reduction	additive avg.	constant	6 random	3.67
rigg	dataset size val. loss integral val. loss reduction	weighted additive avg.	constant	Probab. drop large clients	4.65
HT-TUAS	dataset size parameter distance	additive avg.	constant	4 random	4.69
Flair	multiple grad. descent dataset size	constrained optimization	constant	all	5.85

- For Task 2, the characteristics of the five official submissions were already described in table D2, which was moved to table 3 in the revised manuscript, and extended with more details, including all algorithms evaluated for Task 2.

Table D4: Ranking and characteristics of all algorithms evaluated in Task 2. Four institutions were not used for ranking, as many models could not be evaluated on them due to technical problems. ‘-’ denotes that nothing was reported for this field. Abbreviations: CNN = convolutional neural network, CE = cross-entropy, VAT = virtual adversarial training

Model ID	Rank	Architecture	Loss	Post-processing	Ensemble size	nnU-Net
15	1	U-Net, larger encoder	CE, batch region-based Dice,	ET (small to NCR)	10	yes
35	2	U-Net, larger encoder, multi-scale skip block	focal loss, jaccard, region-based	-	30	no
37	3	U-Net	CE, Dice, TopK, region-based	-	5	yes
38	4	U-Net, residual blocks, transformer in bottleneck	CE, Dice	ET (small to NCR)	3	yes
16	5	U-Net	CE, Dice	ET (drop disconnected), TC (fill surrounded), WT (drop small components)	5	yes
14	6	U-Net, larger encoder	CE, batch Dice, region-based	ET (small to NCR)	5	no
11	7	U-Net	CE, Dice	TC (fill surrounded)	5	yes
54	8	CoTr, HR-Net (CNN), U-Net, U-Net++	multi, region-based	ET (small to NCR)	5	yes + other
10	9	U-Net	CE, Dice, region-based	ET (small to NCR)	5	yes
31	10	U-Net, larger encoder, residual blocks	Dice, focal loss	ET (small to NCR)	5	no
51	11	HNF-Net (CNN), attention	CE, genDice, region-based	ET (small to NCR)	5	no
33	12	U-Net, multiple encoders	CE, Dice, region-based	ET (small to NCR)	4	no
46	13	U-Net	CE, Dice, generalized Wasserstein Dice	-	8	no
40	14	U-Net, larger encoder, residual blocks	Dice, region-based	ET (small to NCR)	4	no
27	15	U-Net, modality co-attention, multi-scale skip block, transformer in bottleneck	CE, region-based	ET (drop small components)	-	no
44	16	U-Net	CE, Dice, region-based	ET (convert to NCR based on auxiliary network), drop small components	10	yes + other
19	17	U-Net	CE, Dice, batch Dice, region-based	ET (small to NCR)	15	yes + other
32	18	U-Net	batch Dice, region-based	ET (small to neighboring label), drop small components	5	no
42	19	-	-	-	-	-
18	20	HardNet (CNN)	CE, Dice, focal loss, region-based	-	3	no
48	21	U-Net, attention	Dice, region-based	-	1	no
25	22	U-Net, attention	CE, Dice, region-based	-	1	no
13	23	-	-	-	-	-
26	24	U-Net, multiple decoders	CE, Dice, region-based	TC (remove outside of WT), drop small components, morph. closing	1	no
30	25	2-stage, 2D, CNN, U-Net, U-Net++, residual blocks	Dice	-	29	no
41	26	CNN, neural architecture search	CE, Dice, region-based	-	5	no
8	27	Swin Transformer	CE, Dice, VAT, region-based	-	1	no
12	28	U-Net	Dice, region-based	-	1	no
47	29	U-Net	CE, Dice	-	1	no
22	30	2D, U-Net, attention, residual blocks	CE, Dice	unknown	-	no
45	31	2-stage, U-Net, residual blocks	CE, Dice, region-based	ET (small to NCR)	5	no
52	32	U-Net, attention, residual blocks	Dice, region-based	-	5	no
36	33	2D, U-Net, residual encoder	Dice	-	1	no
23	34	2D, U-Net, residual encoder, transformer	CE, Dice, region-based	-	1	no
39	35	2-stage, U-Net	unknown	-	1	no
43	36	U-Net, multi-stage	BCE	fill holes	1	no
21	37	2D, U-Net++	Dice, boundary distance	-	3	no
28	38	2-stage, CNN, Graph NN	CE	-	1	no
53	39	CNN, larger encoder, residual blocks	Dice, boundary, region-based	ET (small to NCR)	1	no
29	40	2D, U-Net	Dice	-	1	no
24	41	-	-	-	-	-

[R2.5.1] Pati, Sarthak, et al. "Federated learning enables big data for rare cancer boundary detection." Nature communications 13.1 (2022): 7346.

6. Related to point 5, it seems that for authors are evaluating mainly aggregation weights strategies leading to variants of federated averaging. While this setup is possible this disregards most of the advances in FL algorithms playing on the vectorized updates themselves using improvements like control variates [6] or penalizing shifts from consensus using elastic losses[7] therefore it should be clearly stated that the evaluation is only comparing the aggregation weighting schemes.

RESPONSE:

Indeed, the challenge design focused on methodology for federated weight aggregation and client selection, and did not allow modifying other aspects like the local optimization procedure or model architecture. These constraints were chosen to foster innovation in these specific parts of the FL algorithm and to make performance gains more attributable. As FeTS was historically the first challenge to benchmark FL methods, we also wanted keep the complexity and hence the barrier for participation low. Furthermore, simulating the total FL time becomes increasingly difficult if more degrees of freedom are introduced in the methods.

We agree that this design limitation should be clearly communicated. In the revised manuscript, we made sure that this is adequately conveyed:

- Introduction: We precisely state “Fair comparison of federated aggregation methods” as the primary objective for task 1 and summarize our contribution for this part of the challenge with “The FeTS Challenge Task 1 establishes a standardized evaluation framework for comparing federated aggregation methods”
- Methods: In section 4.4.1 we write “To focus on the development of aggregation methods, we needed a pre-established segmentation model architecture” and in 4.4.2, the exact design space for the challenge participants is described (ll.977 - 987), including “Aggregation method”, “Collaborator selection” and “Hyperparameters for local training”.
- Results: Sections 2.1 and 2.2 specifically focus on two design choices that are reflected in the heading (collaborator selection and weight aggregation).
- Discussion: We added the constraints on the algorithm design as a limitation to the section, following our argumentation from above.

7. In what way does the claim that the challenge "resemble a phase 3 clinical trial" l594 is justified ? Are the authors performing randomization and causal inference ?

RESPONSE:

Our thought was that the large-scale federated evaluation in Task 2 shares some characteristics and goals of a phase 3 clinical trial, “[which] is conducted in a larger and often more diverse target population in order to demonstrate and/or confirm efficacy [...]” [R2.7.1]. [R2.7.2] also compares phases of the development and evaluation of diagnostic algorithms with pharmaceutical trials (their fig. 2 reproduced below), specifically stating that phase III aims “to confirm that the real-world performance of the algorithm matches its performance in the test environment”. There are also differences to a clinical trial. The segmentation algorithm is not integrated in a clinical workflow; hence we do not have a control group or randomization in that context, as the reviewer pointed out.

We added a reference to [R2.7.2] to the introduction and slightly changed the wording of this sentence to make it more precise.

[Figure Redacted]

[R2.7.1] Umscheid, Craig A., David J. Margolis, and Craig E. Grossman. "Key concepts of clinical trials: a narrative review." Postgraduate medicine 123.5 (2011): 194-204.

[R2.7.2] Larson, David B., et al. "Regulatory frameworks for development and evaluation of artificial intelligence–based diagnostic imaging algorithms: summary and recommendations." Journal of the American College of Radiology 18.3 (2021): 413-424.

Minor

1. white tiles from Fig 2 indicate missing runs due to "issues related to GPU compatibility" I735. Why couldn't the python script fall back on CPU if that was the case ? Couldn't the docker image protocol be robustified ?

RESPONSE:

The compatibility issues only affected the BraTS submissions from the previous year. For these, CPU inference was not feasible within the available timeframe, which was limited by the availability of collaborators and their hardware resources. While a fallback option to CPU inference is in principle possible and a good idea, it also depends on the original submission code whether this works. If GPU usage is hard-coded in the inference code by participants, for example, the docker would still fail. Dockers supporting both CPU and GPU inference can be a requirement for new challenges, but for the 36 "imported" BraTS submissions from the previous year it is not guaranteed that CPU mode will work. For the 5 official FeTS challenge submissions, GPU compatibility was not a problem, because we provided precise instructions for how to avoid related issues.

2. Fig 2. heatmap is not well-chosen as the spectrum between dark colors indicating "bad" performance and lighter colors is difficult to follow the reviewer advises the authors to use other colormaps potentially non continuous if it is simpler to see i.e. from 0.5 to 0.7 DICE: dark red, from 0.7 to 0.8 orange, from 0.8 to 1. green.

RESPONSE:

In the original, perceptually uniform color map, a potential lack of robustness (i.e., lower DSC) will be highlighted as red/darker hues. We acknowledge that the colors are not very intuitive and switch to a color palette ranging from red (bad performance) over yellow to green (good performance), as suggested by the reviewer.

3. The reviewer would appreciate seeing in Fig3. how do other models fare ? Is performance ordered similarly across institutions for different models ?

RESPONSE:

In figure 3, we wanted to show that there are failure cases on many institutions, independently of whether they were seen during training or not. We also wanted to describe the metric distribution for each site in more detail, with metric values on a single-case level (these were "averaged out" in fig. 2). To avoid making the figure overly complicated, we focused only on the overall best model and argue that similar findings are applicable for all models.

Fig. 2 already shows that the average performance between models is similar. To make this argument even more convincing we thought about ways to compare other models' performance, as suggested by the reviewer. One option is to plot for each institution (x-axis) the distribution of "average case model performance" for each model (e.g., median DSC across all test cases) and the "worst case performance" for each model (e.g., 10th percentile of DSC across all test cases). The resulting figure shows that the median performance is very similar between models. For the "worst-case" performance, the differences are bigger, but the overall trend is still comparable for most institutions. Hence, this plot confirms that failure cases occur for most institutions, independently of whether they were seen during training, and that this trend is consistent between different models.

We suggest adding this visualization to the supplementary material (fig. C8) and referencing it in the results section 2.3. Furthermore, we stress in this section that the trend between models is similar.

4. Aggregation schemes used by participants should be described in details in Methods. Descriptions like "together with a sophisticated method for selective sampling that models centers using a Poisson distribution" l664 are insufficient to assess the methods even if results can be found in referenced technical reports.

RESPONSE:

We agree that the teams' methods description was insufficient and clarified important details. To keep each description concise, we avoided including formulas, but would like to hear the reviewer's opinion whether the current version is detailed enough or whether it should further be refined in a more mathematical presentation.

- [1] Pati, S., Baid, U., Edwards, B. et al. Federated learning enables big data for rare cancer boundary detection. *Nat Commun* 13, 7346 (2022). <https://doi.org/10.1038/s41467-022-33407-5>
- [2] Bujotzek, Markus R., et al. "Real-World Federated Learning in Radiology: Hurdles to overcome and Benefits to gain." arXiv preprint arXiv:2405.09409 (2024).
- [3] Saldanha, O. L., Quirke, P., West, N. P., James, J. A., Loughrey, M. B., Grabsch, H. I., ... & Kather, J. N. (2022). Swarm learning for decentralized artificial intelligence in cancer histopathology. *Nature medicine*, 28(6), 1232-1239.
- [4] Kairouz, P., McMahan, H. B., Avent, B., Bellet, A., Bennis, M., Bhagoji, A. N., ... & Zhao, S. (2021). Advances and open problems in federated learning. *Foundations and trends® in machine learning*, 14(1–2), 1-210.
- [5] Sattler, F., Wiedemann, S., Müller, K. R., & Samek, W. (2019, July). Sparse binary compression: Towards distributed deep learning with minimal communication. In 2019 International Joint Conference on Neural Networks (IJCNN) (pp. 1-8). IEEE.
- [6] Karimireddy, Sai Praneeth, et al. "Scaffold: Stochastic controlled averaging for federated learning." *International conference on machine learning*. PMLR, 2020.
- [7] Li, T., Sahu, A. K., Zaheer, M., Sanjabi, M., Talwalkar, A., & Smith, V. (2020). Federated optimization in heterogeneous networks. *Proceedings of Machine learning and systems*, 2, 429-450.
- [8] Caldas, Sebastian, et al. "Leaf: A benchmark for federated settings." arXiv preprint arXiv:1812.01097 (2018).

Response to the reviewers

The authors would like to thank the reviewers for their positive response to our first revision, and the additional suggestions for making the wording of the multi-site evaluation more precise, removing unnecessary figures and improving the published code.

We addressed all major and minor points raised by the reviewers in our second revision. Please see our responses below, where we also describe how we revised the manuscript and code based on the reviewer's feedback

As before, for the reviewers' convenience, we have highlighted significant changes in the PDF with the revised manuscript in **red**.

Reviewer 1

Reviewer #1 (Remarks to the Author):

I have carefully read through the response from the reviewers, and would like to thank the authors for such effort.

I would say, the authors have resolved all my concerns.

RESPONSE:

We thank the reviewer again for the valuable feedback, without which this manuscript could not have reached the current quality.

Reviewer 2

Reviewer #2 (Remarks to the Author):

Overall the reviewer considers the authors have answered their main concerns and that the manuscript is in a much better shape now. In particular the reviewer appreciates the effort spent in notably 1. reformulating the abstract to emphasize that FL is simulated as well as 2. highlighting the differences between the tasks and 3. explaining the method to compute "communication efficiency" in details.

However, the reviewer still feels very uncomfortable with the "federated evaluation" wording and would like the authors to change it. As for the exact wording to be used, the

authors themselves use "collaborative, multi-site evaluation" l558 and inference in the wild" l1073 which are much better suited and the reviewer is fine with both formulations. In order to emphasize the contribution of this paper on the reproducibility and harmonization of the evaluation protocol, which is significant, the reviewer would accept any expression such as "reproducible multi-site evaluation protocols", expressions including adjectives such as "fair" or even acronyms to facilitate messaging depending on the authors' inclinations as long as they do not include the word "federated".

The reviewer would like to point out that this attention to details on the wording is not simply a reviewer's whim. Federated Learning for medical research is a complex multi-disciplinary field where both transparency and attention to messages and wording is of utmost importance so that medical practitioners and partner institutions' representatives become aware of the challenges and the cost that it currently entails.

Nowadays there is widespread misunderstanding of both the true privacy guarantees offered by FL as well as the difficulty of actually setting up such non-simulated collaborations in practice.

As a part of the community, the reviewer takes some part of responsibility for this current state of things but encourage the authors to participate with the reviewer in the effort to change mentalities by being crystal clear when describing what they are doing.

This article illustrates very nicely such challenges and it is the reviewer's hope that this contribution, alongside like-minded initiatives, will help bring the community closer of true reproducible standardized federated trainings between partners institutions. However, as the paper shows, currently even the multi-site evaluation step and siloed training represents already a huge challenge that the authors tackle without even considering non-simulated federated learning, which is much trickier.

RESPONSE:

We completely agree that real-world federated learning projects are still a challenging endeavor with many technical and practical hurdles to overcome. From this perspective we can understand that the wording "federated evaluation" could be too vague and make it harder for readers to grasp the significant step from a collaborative, multi-site evaluation to federated learning.

Based on the previous discussions and the reviewer's suggestion, we decided to replace all occurrences of "federated evaluation" with "collaborative, multi-site evaluation", or sometimes shorter "multi-site evaluation" if the context is clear. We refrained from adding further adjectives or using acronyms, as this would inflate the expression for such a simple but powerful concept unnecessarily.

Changes in the manuscript: All sections and figures were revised such that (collaborative) multi-site evaluation is used instead of federated evaluation. The abstract was also slightly rearranged and trimmed to reduce the word count.

Minor:

- The computation of the "communication efficiency" is now much clearer however it is very mysterious to the reviewer how gaussian sampling helps evaluation or ranking. Could the authors clarify that ?

RESPONSE:

Randomizing the federated round timings for each client in Task 1 was not done to help evaluation or ranking, but rather to increase the realism of the federated learning simulation. During a previous study, the mean and standard deviation of individual, fluctuating timings (training time per case, validation time per case, model download/upload time) were measured for each participating institution in a real-world federated learning project that trained a similar network as Task 1 [Pati22]. By incorporating these timings with realistic random noise in the FeTS Challenge, our goal was to simulate realistic FL conditions and enhance the practical relevance of challenge submissions.

While the motivation for the Gaussian sampling is not related to evaluation/ranking, perhaps the reviewer was thinking about how evaluation could be affected by the randomness. In that regard, we would like to stress that all FL algorithm submissions were trained using the same round duration values, to ensure a fair comparison and reproducibility. This was achieved by using a fixed random seed and making sure that one value is sampled per collaborator in each round.

In conclusion, although the sampling is not strictly necessary, it made the FL simulations more realistic. We updated the description in section 4.3.2 to explain this background better.

References:

[Pati22] Pati, Sarthak, et al. "Federated learning enables big data for rare cancer boundary detection." *Nature communications* 13.1 (2022): 7346.

Extra details:

Most standard deviations are small (<10% of mean duration). The exact timings can be found in

<https://github.com/FeTS->

[AI/Challenge/blob/2c3e2cc01d23af936567ad6c602dda9dc2f87192/Task_1/fets_challenge/experiment.py#L33](https://arxiv.org/abs/2003.01554v1)

- Another issue currently present in the article is the lack of details in the Methods especially on each team's contributions. Indeed as asked by the authors in their rebuttal to comment m2.4: it is not blocking but yes, the reviewer would like preferably a more precise description of at least the aggregation methods used by the different teams using mathematical formulas. The reviewer advise the authors to not shy away from throwing a few equations in this very practical article as it would help comprehension a great deal and the reviewer expects readers to gain lots of insights by reading the details of say "FedPIDAvg" or "Adam as a server optimizer" which are too vague. The reviewers understands there are many teams and it is time consuming to do the back and forth but potentially it could be done only for a selection of say the top 3 most innovative aggregation methods.

RESPONSE:

We understand that some readers may prefer a short mathematical summary of the submitted algorithms over looking up the original paper, so we added concise formulas of the aggregation weighting terms for the task 1 algorithms. For the team FLAIR, a precise mathematical description is out of scope for this document, because it departed considerably from the "FedAvg way". As this method did not work very well in their experiments and was published before, we concluded that adding extensive mathematical background on this is not worthwhile. For team RoFL, the concept behind the "server optimizer" (from a previous publication) was explained, but the details of the Adam optimizer left out, as is common for most papers using it. While the new formulas explain the weighting terms p , note that the meta-approach of additive/multiplicative averaging of multiple weighting terms was already added in the previous revision. Further, a description of Task 2 algorithm characteristics mentioned in table 3 were added in the "additional information on Task 2 algorithms" subsection, to clarify keywords and insert references for methods that cannot be explained in the scope of this manuscript. We hope that the reviewer finds the additions useful for clarifying important details.

- Fig 6 is not bringing anything to what Fig 1 already presents. Furthermore it is a reproduction from another article, which is a bit lazy. Considering Figure 5, The reviewer would like the U-net structure to be put in perspective using the radiology segmentation task that is studied here at least and potentially integrated with Fig 6 if necessary to emphasize the federation part (although Figure 1 is already quite clear). Maybe emphasize the specificities of nn-Unet and the fact that most participants chose the same architecture ? As a recap, neither Figure 5 nor Figure 6 seem to bring anything to the article in their current state. The reviewer would like them to be either improved/replaced by better figures from the supplementary or removed.

RESPONSE:

We agree with the reviewer that the current figures do not contribute in the manuscript, and therefore followed the reviewer's suggestion and removed them from the revised version.

- Even though the reviewer likes the new wording used to compare this work with clinical trials. The reviewer would like that comparison removed entirely as the current evaluation protocol is nowhere near achieving the standards of clinical trials.

RESPONSE:

The comparison with clinical trials was removed from the manuscript.

Reviewer #2 (Remarks on code availability):

From a very very quick pass through the repository <https://github.com/FETS-AI/Challenge>.

Positive points:

- There are lots of detailed instructions, READMEs, documentations and links to websites to make a submission
- it uses docker therefore implies some reproducibility but through mlcube, which is an abstraction the reviewer is not familiar with
- Q&A forums accessible
- Tasks 1 and 2 correspond to two different distinct folders helping readability

Negative Points

- no tutorial / quickstart clearly accessible from the landing README
- not lightweight/self-contained at all as it uses a mix of R, python code as well as multiple (and somewhat intricate) dependencies
- linked to above points but it seems quite involved to just run some code and will make reproduction of the results quite the endeavor

The reviewer is not sure that the claims in the code reporting summary are supported by the code notably the availability of a small lightweight example to quickly test the code as well as the precise timings on standard hardwares but the reviewer did only a very quick pass and might be mistaken.

RESPONSE:

Providing a well-documented and functional repository was especially important during the challenge to improve the experience of participants. Now, after the challenge, some of the functionality (like submitting a model) is not available anymore, but the website and repository are nevertheless helpful for reproducibility and transparency reasons. Furthermore, future researchers might find such a resource and the public dataset helpful.

In response to the reviewer's comments, we restructured the landing page readme and the task readmes, such that the repository is more self-contained. However, the challenge website (<https://www.synapse.org/#!Synapse:syn28546456>) and repository (<https://github.com/FETS-AI/Challenge>) are designed to complement each other; while the website contains general information about the challenge and how to participate, the repository provides the code templates for participation, as well as essential software components of the multi-site evaluation pipeline for transparency. Both resources continue to be available online for future reference.

The reviewer also criticized the lack of a small, lightweight example, and the difficulty of reproducing results. To address these points, we improved the code repository and clarify below how it can be used to quickly run code and reproduce the paper results.

The source data (evaluation metric values for all evaluated algorithms) underlying all figures and tables in the manuscript will be made available to the public. It is included as source data in the revision files. To quickly reproduce our analysis and ranking, we added a new folder (paper_analysis) to the repository, with instructions how to use it. Note: This part of the repository will only be merged after publication, but is already available online under <https://github.com/mzenk/Challenge/tree/analysis-code-release?tab=readme-ov-file#source-code-for-the-paper-analysis>. We also added a zipped folder with source code to the revision files, along with the required source data files.

The other parts of the repository were meant as a guide for the participants during the challenge. Each provides a runnable example: Task_1 provides detailed instructions on how to modify a simple python script, which implements a simple interface to train models using the challenge's FL framework (link). A baseline algorithm is included in the code, which can be run on a computer without GPU, using a subset (small split) after downloading the publicly available challenge data. The challenge data are not small, but as this code was only meant to be used within the challenge, we think that it is adequate for transparency and as a starting point for further method development.

For Task_2, reproducing the multi-site evaluation on a "standard desktop computer" is unfortunately not possible. We transparently report the tools and pipelines used for this evaluation through the FeTS challenge and the medperf (branch "fets-challenge") repositories. As a demo for how algorithms are evaluated locally, the section "How to run the evaluation pipeline locally" in the Task 2 repo was updated so that it uses fixed python versions. It can be executed on a computer without GPU, because only a dummy model is run. A functional docker installation is required, though. Lightweight example data is included in this demo.

Regarding timings on standard hardware: Running the paper analysis reproduction script takes less than a minute. The FL training on the small split (Task 1) takes about 14h on one of our workstations (without using GPU), while the evaluation pipeline test run for Task 2 takes about 1:30min, plus additional time for downloading docker containers (~34GB). We added this information to the repository.

Changes in the manuscript: we added additional links to the challenge website and the medperf repository branch used for the FeTS challenge in the code availability section. The source data publication was added to the data availability section.